

# The transcriptomic response to a short day to long day shift in leaves of the reference legume *Medicago truncatula*

Geoffrey Thomson, James Taylor and Joanna Putterill

School of Biological Sciences, University of Auckland, Auckland, New Zealand

## ABSTRACT

Photoperiodic flowering aligns plant reproduction to favourable seasons of the year to maximise successful production of seeds and grains. However understanding of this process in the temperate legumes of the Fabaceae family, which are important both agriculturally and ecologically, is incomplete. Previous work in the reference legume *Medicago truncatula* has shown that the *FT-like* gene *MtFTa1* is a potent floral activator. While *MtFTa1* is upregulated by long-day photoperiods (LD) and vernalisation, the molecular basis of this is unknown as functional homologues of key regulatory genes present in other species, notably *CONSTANS* in *A. thaliana*, have not been identified. In LD *MtFTa1* maintains a near constant diurnal pattern of expression unlike its homologue *FT* in *A. thaliana*, which has a notable peak in expression at dusk. This suggests a different manner of regulation. Furthermore, *M. truncatula* possesses other *FT-like* genes such as two LD induced *MtFTb* genes which may also act in the regulation of flowering time. *MtFTb* genes have a diurnal pattern of expression with peaks at both four and sixteen hours after dawn. This study utilises RNA-Seq to analyse the transcriptome of *M. truncatula* leaves to identify genes which may regulate or be co-expressed with these *FT-like* genes following a shift from short-day photoperiods to inductive long-days. Specifically this study focuses on the first four hours of the day in the young leaves, which coincides with the first diurnal peak of the *FTb* genes. Following differential expression analysis at each timepoint, genes which alter their pattern of expression are distinguished from those which just alter their magnitude of expression (and those that do neither). It goes on to categorise these genes into groups with similar patterns of expression using c-means clustering and identifies a number of potential candidate photoperiod flowering time genes for future studies to consider.

## INTRODUCTION

The regulation of flowering controls the important developmental shift between the vegetative and reproductive growth phases of the plant, aligning plant sexual reproduction with favourable seasonal environmental variation. This facilitates successful pollination and the maximizing of crop productivity and yield. In many temperate climate species, such as the winter annual varieties of the well-studied Brassicaceae species *Arabidopsis thaliana* (L.) Heynh., the primary determinants of flowering time are long-day (LD) photoperiod

Corresponding authors
Geoffrey Thomson,
gtho123@aucklanduni.ac.nz
Joanna Putterill,
j.putterill@auckland.ac.nz

(daylength) conditions and vernalisation (prolonged exposure to cold temperatures) for which the molecular pathways underlying these responses are well understood (*Andrés & Coupland, 2012*).

In *A. thaliana*, LD conditions are perceived in the leaves and culminates in the activation of the floral integrator gene *FLOWERING LOCUS T* (*FT*) (*Turck, Fornara & Coupland, 2008*). Specifically, *FT* is activated by circadian and light signals aligning which facilitates the formation of the GIGANTEA-FLAVIN-BINDING KELCH REPEAT, F-BOX 1 (GI-FKF1) complex. This complex degrades CYCLING DOF FACTOR (CDF) transcription factors which otherwise form a complex with a TOPLESS (TPL) protein to repress the expression of the transcription factor *CONSTANS* (*CO*). This gene encodes a B-box class protein with a CCT domain (*Putterill et al., 1995*; *Goralogia et al., 2017*). The stabilisation of CO protein in the late afternoon of LD results in it acting as a subunit of a NUCLEAR FACTOR-Y (NF-Y) pioneer transcription factor complex which directly activates *FT* (*Andrés & Coupland, 2012*; *Gnesutta et al., 2017*). CDF proteins are also able to repress *FT* directly (*Song et al., 2015*). FT protein is the principal mobile floral signal (florigen) which is transported to the shoot apical meristem via the phloem to induce flowering via activation of a second floral integrator gene *SUPPRESSOR OF OVEREXPRESSION OF CONSTANS 1* (*SOC1*) and the floral meristem identity gene *APETALA1*. This results in a state of floral commitment and the development of flowers (*Turck, Fornara & Coupland, 2008*).

Beyond *A. thaliana*, the presence of *FT* orthologues integrating environmental signals and regulating flowering appears to be widely conserved (*Ballerini & Kramer, 2011*; *Wickland & Hanzawa, 2015*; *Putterill & Varkonyi-Gasic, 2016*). However the conservation of other elements of the pathway is not as strong. Recent work by *Simon et al. (2015)* suggest that the central role *CO* plays in the photoperiod pathway in *A. thaliana* evolved within the Brassicaceae following a gene duplication. Thus *CO-like* (*COL*) genes regulating *FT* homologues is not universal and their activity in some other species (e.g., *Oryza sativa* L.; rice; *Hayama et al., 2003*) are the result of convergent evolution. Examples of species in which *COL* genes do not appear to regulate flowering time include Japanese morning glory (*Ipomoea nil* (L.) Roth) and temperate Fabaceae family (legume) species such as pea (*Pisum sativum* L.) and *Medicago truncatula* Gaertn. Notably, a recent survey of *COL* genes in *M. truncatula* found no evidence of them acting as a photoperiodic switch (*Wong et al., 2014*) and the gene expression of the closest *COL* homologue in pea, was not altered in photoperiodic flowering time mutants. These include the recessive *late bloomer1* (*late1*) mutant which disrupts the orthologue of *GI* (*Hecht et al., 2007*; *Liew et al., 2009*) and a dominant late flowering mutant *late2* which has been mapped to a *CDF* gene *PsCDFc* (*Ridge et al., 2016*). This *Pscdfc* mutant encodes a protein which has lost the ability to interact with the FKF1 and exhibits reduced expression of *FT-like* genes (*Ridge et al., 2016*). Whether the regulation of the *PsFT-like* genes by PsCDFc is direct or not remains unknown.

Nevertheless, in many species the regulation of *FT* orthologues exhibit significant commonalities such as genes containing CCT and/or B-box domains integrating the photoperiod and vernalisation signals. For instance, in the Pooid grasses such as wheat (*Triticum spp.*) prior to vernalisation the repression of the *FT-like* gene is maintained by a pair of ZCCT proteins, which contain both CCT and B-box domains, in complex

with NF-Y subunits (*Song et al., 2015*; *Li et al., 2011*). Then in inductive LD conditions upregulation of the *FT* orthologue requires the *PHOTOPERIOD 1* (*PPD1*) gene, which encodes a CCT domain (*Shaw et al., 2013*; *Pearce et al., 2017*). In addition genes encoding CCT and B-Box domains are important in the cultivated varieties of the LD responsive sugar beet (*Beta vulgaris* L.) (*Pin et al., 2012*; *Dally et al., 2014*).

The lack of a direct upstream regulator of *FT-like* genes in temperate legumes means that, despite good progress, the understanding of how flowering time in this family is incomplete (*Putterill et al., 2013*; *Weller & Ortega, 2015*). Legumes are an ecologically diverse plant family (*The Legume Phylogeny Working Group, 2013*) and include a number of dietary staple crops. In an agricultural context these crops reduce the need for fertilizer use via nitrogen fixation (*Vance, 2001*). Like *A. thaliana*, many temperate legume species, including *M. truncatula*, accelerate their flowering in response to vernalisation and LD conditions (*Highkin, 1956*; *Summerfield et al., 1985*; *Roberts, Hadley & Summerfield, 1985*; *Laurie et al., 2011*; *Weller & Ortega, 2015*; *Ridge et al., 2017*). Classically pea has been most intensively studied to analyse photoperiodic flowering in temperate legumes but has recently been complemented by the study of other species, such as *M. truncatula* for which considerable genetic and genomic resources exist (*Tadege et al., 2008*; *Young et al., 2011*; *Tang et al., 2014*).

Analysis of flowering time mutants in pea has demonstrated that some members of the photoperiod pathway, such as the photoreceptors and components of the circadian clock, are largely conserved with *A. thaliana* (*Weller & Ortega, 2015*). In addition, *FT-like* homologues have been characterised in several legume species with most having multiple copies which fall into three sub-clades (*Laurie et al., 2011*; *Hecht et al., 2011*). In pea and *M. truncatula* the *FT-like* gene *FTa1* is a potent floral inducer whose expression is elevated in LD (*Laurie et al., 2011*; *Hecht et al., 2011*). However unlike *FT* in *A. thaliana*, *MtFTa1* (*Medtr7g084970*) does not possess a diurnal pattern and instead exhibits a near constant level of expression once induced in LD (*Laurie et al., 2011*) suggesting that the mechanisms of regulation between *FT* and *FTa1* likely differ significantly. Moreover grafting experiments in pea between flowering time mutants suggest that additional floral stimuli exist (*Hecht et al., 2011*).

Good candidates for secondary floral stimuli are the *FTb* genes that, like *FTa1*, are upregulated in LD and capable of complementing an *A. thaliana ft-1* mutant (just *MtFTb1* in *M. truncatula*, *Laurie et al., 2011*; *Hecht et al., 2011*). Distinctively, *MtFTb* genes have a diurnal pattern of expression in LD and peak twice, at ZT4 and ZT16 (ZT is zeitgeber time where ZT0 is subjective dawn at lights on). This pattern of expression is similar to that of *FT* in *A. thaliana* under "natural" LD conditions (*Song et al., 2018*). Another legume *FT-like* gene which may play a role in flowering time is *FTa2* which in *M. truncatula* it is mostly expressed in short-day (SD) photoperiod conditions consistent with a floral repressor (*Laurie et al., 2011*). However whether any *FT-like* genes other than *FTa1* regulate flowering time in either pea or *M. truncatula* remains to be demonstrated.

Downstream of *FT-like* genes the regulation of flowering in temperate legumes is similar overall to that of *A. thaliana*, albeit complicated by several genes being present in multiple copies with potential functional redundancies. For example, three *MtSOC1-like*

genes depend on *MtFTa1* for the extent and timing of their expression, although *Mtsoc1a* mutants do show delayed flowering (*Fudge et al., 2018*; *Jaudal et al., 2018*). The genes involved in inflorescence development are similar to that of *A. thaliana* (*Cheng et al., 2018*).

Overall, while some components of the *A. thaliana* photoperiodic flowering model appear to be conserved in temperate legumes, other aspects differ. Specifically, what factors act immediately upstream of *FT-like* genes in the photoperiodic flowering of temperate legumes remain unknown. In light of the gap in understanding, we take a transcriptomic approach to identify additional candidate regulators of photoperiodic flowering in *M. truncatula*. In two experiments we target genes expressed in a similar, or opposite, manner to LD induced *FT-like* genes with the aim of identifying candidate regulators. Plants were shifted from SD (8 h light/16 h dark) to LD (16 h light/8 h dark) conditions and gene expression changes analysed in the first four hours of the diurnal cycle; at dawn (ZT0), two hours after dawn (ZT2) and four hours after dawn (ZT4) during which time the LD induced *FT-like* genes are expressed. These timepoints capture the constant induction of *MtFTa1* in LD and target the first diurnal peak of *MtFTb1* and *MtFTb2* at ZT4 also in LD (*Laurie et al., 2011*).

## MATERIALS AND METHODS

### Growth of plants and tissue sampling

*M. truncatula* cv 'Jester' (*Hill, 2000*) seeds were scarified by softly scraping them between two pieces of sand paper (grade P160). The seeds were then germinated at 15 °C in gently shaking tubes of water and dark conditions for 24 h. Germinated seeds were then vernalised by being transferred to damp petri dishes and incubated at 4 °C for a further 25 days. The seedlings were subsequently planted in seed raising mix (Daltons Ltd., NZ) in individual cell pots and grown in growth cabinets at 22 °C under 240 $\mu$Mm$^{-2}$sec$^{-1}$ cool white fluorescent light. This was in accordance with Institutional Biological Safety Committee approval GMO08-UA006. Soil was kept moist with a complete liquid nutrient media (without $Na_2SiO_3$; *Gibeaut et al., 1997*). In the two experiments plants were grown in SD conditions (8 h light/16 h dark) until they were 10 days old and were then shifted at ZT8 into LD conditions (16 h light/8 h dark) for 3 days in experiment 1 (harvest at ZT0 and ZT2) or 5 days in experiment 2 (harvests at ZT4). Three biological replicates were taken each consisting of two pooled trifoliate leaves from different non-adjacent plants giving a total of 18 samples. Only the first trifoliate leaf to unfurl was sampled from a given plant. Samples were immediately frozen in liquid nitrogen.

### RNA extraction and sequencing

The 18 frozen leaf samples were ground using five 3 mm sterilised ball bearings per sample and a Geno/Grinder® 2010 (SPEX® SamplePrep, USA). Total RNA was then extracted from the ground samples using the RNeasy Plant Mini Kit (Qiagen, Hilden, Germany) following the manufacturer's instructions and the quantities and qualities of the extracted RNA were then measured using a Bioanalyzer 2100 (Agilent Technologies, Santa Clara, CA, USA). Samples were then sent to the Otago Genomics Facility (www.otago.ac.nz/genomics/index.html) and RNA-Seq libraries were prepared.

The first experiment (ZT0 and ZT2 samples) used the ScriptSeq Complete Kit (Plant) (100 bp reads; Illumina Inc., USA) while the second experiment (ZT4 samples) used TruSeq Stranded mRNA libraries (120 bp reads; Illumina Inc., USA). Each experiment was sequenced on a single lane of a HiSeq2000 (Illumina Inc., USA).

## Complementary DNA synthesis and RT-qPCR analysis

Following RNA extraction, 8 μg of RNA was treated with the TURBO DNA-free Kit (Invitrogen, Foster City, CA, USA) . First-strand complementary DNA (cDNA) was then synthesised using 1 μg of DNase treated RNA using SuperScript IV Reverse Transcriptase (Invitrogen, USA) using an oligo dT primer following the manufacturers instructions. At this point a control reaction where the reverse transcriptase is omitted was run, one reaction per set of replicate samples. When tested alongside synthesised cDNA these reactions control for the presence of genomic DNA contamination.

Measurement of relative abundances of cDNA, as a measure of gene expression, was done using Real time quantitative PCR (RT-qPCR). In this assay 10 μl reactions using Power SYBR Green PCR Master Mix and 2 μl of diluted cDNA (cDNA was diluted 20x prior to RT-qPCR). Control reactions using water, as well as the cDNA reactions which lacked reverse transcriptase, were run simultaneously. RT-qPCR experiments were assembled on 384-well plates and run on an Applied Biosystems 7900HT Sequence Detection System.

Analysis of the RT-qPCR data was done using the $2^{-\Delta\Delta Ct}$ algorithm (*Livak & Schmittgen, 2001*). This utilised either *Medtr7g089120* a *TUBULIN BETA-1 CHAIN* gene or *Medtr6g084690* a *SERINE/THREONINE PROTEIN PHOSPHATASE 2A REGULATORY SUBUNIT* (*PP2A*; previously known as *PROTODERMAL FACTOR 2*) as reference genes (*Kakar et al., 2008*). Primers used are available in Table S2.

## RNA-Seq analysis

Read trimming of the the FASTQ files was then performed using BBDuk tool in the BBTools suite (v37.54; *Bushnell, 2018*). This removed the sequencing adapters and low quality sequence (Phred = 20) and retained only reads which were at least 36 bases in length. Furthermore read pairs lacking one of the pair were discarded. Transcripts were then quantified using Salmon (v0.8.2; *Patro et al., 2017*) which uses quasi-mappings to map the reads to the annotated genes of the Mt4.0v2 transcriptome (*Young et al., 2011*; *Tang et al., 2014*). The resulting count tables were then imported into R (*R Core Team, 2018*) using the tximport package (v1.4.0; *Soneson, Love & Robinson, 2015*) and principal component analysis (PCA) and DE analysis at the gene level was performed using DESeq2 (v1.16.1; *Love, Huber & Anders, 2014*). DESeq2 normalizes the data by fitting a negative binomial GLM with a gene-specific dispersion parameter. Clustering was performed using the Mfuzz package (v2.36; *Kumar & Futschik, 2007*) using minimum centroid distances as a heuristic measure of appropriate cluster number. Briefly this consisted of compromising between cluster size and number by ascertaining when additional clusters only marginally increased the resolution.

## Data processing and visualisation

All computation and analysis was done on a Macbook Pro (2012; Intel® Core™ i7-3520M upgraded to 16 Gb RAM) from Apple Inc. (Cupertino, CA, USA) running Antergos Linux (v17.12; 4.14.11-1-ARCH kernal). Data processing and visualisation was done in R (*R Core Team, 2018*) using the Tidyverse suite of packages (v1.2.10; *Wickham, 2017*) with additional visualisation using the UpsetR package (v1.4; *Conway, Lex & Gehlenborg, 2017*) and the Superheat package (v0.1; *Barter, 2018*). Analysis and graphs can be reproduced from the accompanying collection of scripts and files in an accompanying figshare repository (see *Thomson, 2018*).

## RESULTS

It has previously been demonstrated that shifting vernalised *M. truncatula* plants from SD to LD induces flowering and is accompanied by the induction of the expression of *FT-like* genes *MtFTa1*, *MtFTb1* and *MtFTb2*. It was found that three days in LD was sufficient to promote the transition to flowering in *M. truncatula* (but one was insufficient; *Laurie et al., 2011*). We thus utilised at least three days in LD in our experiments. Here two similar experiments are analysed where plants were grown in SD conditions until they were 10 days old and were then shifted at ZT8 into LD conditions to describe the transcriptomic changes which occur in the *M. truncatula* leaves following such shifts, alongside the *FT-like* genes.

In the first experiment sampling of leaves occurred at ZT0 (subjective dawn) and ZT2 when the plants had experienced three days of LD conditions. In the second experiment the plants had experienced five days of LD conditions and sampling occurred at ZT4. With the aim of identifying candidate regulators of *FT-like* genes these samples capture the constant LD induction of *MtFTa1* and both precede and include the first diurnal peak of *MtFTb1* and *MtFTb2* at ZT4 in LD (*Laurie et al., 2011*). These samples (in triplicate) were used to construct RNA-Seq libraries which were subsequently sequenced.

The sequenced RNA-Seq libraries all generated 40-50 million reads with mean quality scores greater than 35. The quantification of gene abundances reported an average mapping rate of 89.86% to the Mt4.0v2 transcriptome (see Table S1 for full table of abundances in Transcripts per Million; TPM). Thus the data generated are of a high quality and indicates that the Mt4.0v2 transcriptome is reasonably complete.

### Differential expression at each time point

Analysis of this data initially considered pairwise comparisons of gene expression between LD and SD at each timepoint. The significance of any differences observed was assessed using Wald significance tests and since there are three sets of tests, the significance levels of these were adjusted for together using the false discovery rate method.

It was observed that 28,151–29,011 genes had read counts >1 (out of the 50,444 in the Mt4.0v2 transcriptome). Of these 6,824 genes at ZT0 (24% of expressed genes) had statistically different expression ($\alpha = 0.05$). There were 5,523 genes (19%) at ZT2 and 7,743 genes (25%) at ZT4 (Table S3). When these lists were filtered for those which had >2-fold differences and had >10 mean normalised reads then 2,436, 1,309 and 2,661 genes were

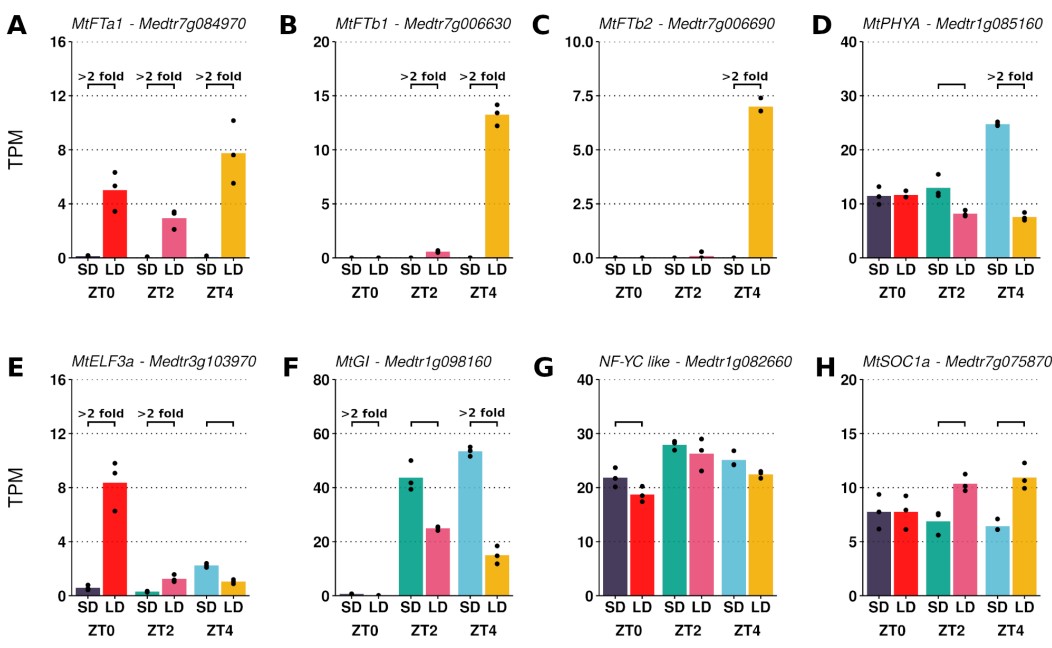

**Figure 1** **Pairwise comparisons of photoperiod induced changes in expression between LD and SD of a selection of candidate photoperiod pathway genes.** Graphs show the mean transcript abundance (in TPM) of (A) MtFTa1, (B) MtFTb1, (C) MtFTb2, (D) MtPHYA, (E) MtELF3a, (F) MtGI, (G) the NF-YC-like gene (Medtr1g082660) and (H) MtSOC1a across the three timepoints. Points are the individual replicate libraries. Statistically significant differences ($\alpha = 0.05$) are indicated by the bracket with those differences which show at least a 2-fold change in transcript abundance annotated.

judged to be DE at ZT0, ZT2 and ZT4 respectively. These numbers are similar to what was observed in an *A. thaliana* microarray experiment after a SD to LD shift where 2000 genes were DE (*Wigge et al., 2005*).

The transcript abundances of the LD induced *FT-like* genes and five other candidate photoperiod pathway genes were assessed (Fig. 1). While absent in SD, large increases in *MtFTa1* transcript abundance were observed in LD at ZT0, ZT2 and ZT4. In addition, appreciable expression of *MtFTb1* and *MtFTb2* was only seen in LD at ZT4, with minimal to no expression at ZT0 and ZT2 in either SD or LD.

The differences in expression exhibited by the *FT-like* genes in Figs. 1A to 1C qualitatively agree with previously reported RT-qPCR time course data where in LD *MtFTa1* has an approximately constant level of expression across the day (*Laurie et al., 2011*). *MtFTb1* and *MtFTb2* have been observed to diurnally peak at ZT4 and ZT16 (*Laurie et al., 2011*), consistent with the DE observed here at ZT4 (Figs. 1B and 1C). This indicates that despite originating from different experiments, these datasets could be analysed together as a ZT0-to-ZT4 time series to observe the pattern of gene expression change following a SD to LD shift.

To further demonstrate that these datasets could be analysed together we considered the diurnal expression of four other genes measured by RT-qPCR in an independent ZT0-ZT20 time series experiment and compared them to our RNA-Seq data (Fig. 2). Here
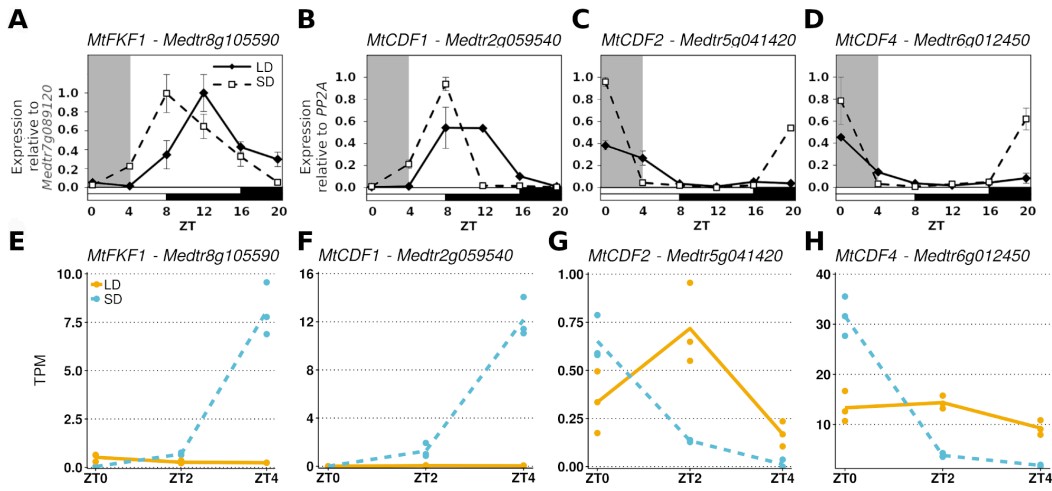

**Figure 2** **Comparing RT-qPCR and RNA-Seq results from independent experiments to validate combining the RNA-Seq datasets.** (A–D) are the diurnal patterns of expression of *MtFKF1* (*Medtr8g105590*), *MtCDF1* (*Medtr2g059540*), *MtCDF2* (*Medtr5g041420* and *MtCDF4* (Medtr6g012450) respectively measured over a diurnal time course using RT-qPCR. The first four hours which overlap with the RNA-Seq data are shaded in grey. Error bars are standard errors of two biological replicates. Samples consist of two fully expanded trifoliate leaves with two biological replicates per time point. For *MtFKF1* these samples were pooled and error bars are standard errors of technical replicates (using *Medtr7g089120* as a reference gene) while for *MtCDF1*, *MtCDF2* and *MtCDF4* these are standard errors of biological replicates (using *PP2A* as a reference gene). (E–H) are the corresponding transcript abundances from the RNA-Seq datasets. The points represent TPM values of the individual replicate libraries plotted with a LOESS smoothed line of best fit for both SD (blue and dotted) and LD (orange and line) samples.

it was observed that in *MtFKF1* (*Medtr8g105590*) and *MtCDF1* (*Medtr2g059540*) from a low point at ZT0 in SD gene expression increases and peaks at ZT8 (Figs. 2A and 2B). This is also seen in the gene abundances of the RNA-Seq datasets where a large increase in SD at ZT4 is observed. Similar minimal LD expression at these timepoints between the two datasets is also in evidence (Figs. 2E and 2F). Conversely, in the RT-qPCR time course *MtCDF2* (*Medtr5g041420*) and *MtCDF4* (*Medtr6g012450*) have their greatest expression at ZT0 and which in SD sharply decreases at ZT4 (Figs. 2C and 2D) which is also observed in the RNA-Seq transcript abundances (Figs. 2G and 2H).

### *Overview of the time series analysis*

The similarity between the patterns of expression observed in this data with previously reported RT-qPCR results for three genes (Figs. 1A to 1C), as well as independently collected RT-qPCR results for an additional four genes (Fig. 2) indicates that there is no significant batch effect which would bias the interpretation of this data as a time series. We then analysed the data in such a manner.

Across the three timepoints 31,363 genes (out of the 50,444) had a read count >1 at at least one timepoint (see Fig. S1 for dispersion plot and MA-plot) and were included in the analysis. To take an overarching view of the variation between samples, principal component analysis (PCA) was employed with the first two principal components plotted in Fig. 3. Biological replicates clustered together and 64% of the observed variation is

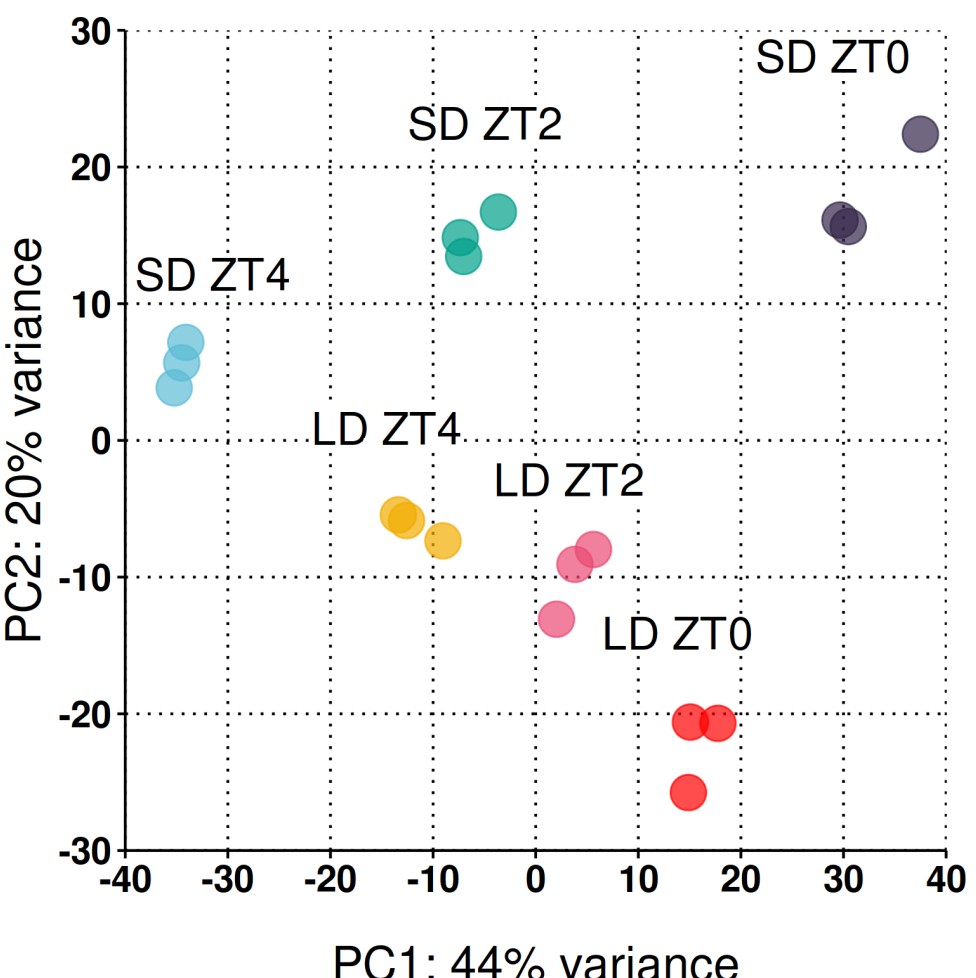

**Figure 3 Reduced-space plot of the first two components of a PCA of individual RNA-Seq libraries.**
This plot gives an overarching perspective on the variation within the dataset with the first two components explaining 64% of the variation. PC1 strongly aligns with the time of sampling and PC2 the photoperiod condition. The plot was constructed using $\log_2$ normalised counts of genes with non-zero read counts.

explained by the first two principal components which strongly align with the time of sampling (PC1) and the photoperiod condition (PC2).

Typically in a pairwise comparison, differences in transcript abundances can be filtered based on fold-change and levels of expression to focus on genes more likely to be biologically consequential. This is difficult in a time series as it is unknown in this instance which timepoint is most relevant to the regulation of *FT-like* genes. Reflecting on the gene abundances plotted in Fig. 1 it was observed that the genes with differing expression between LD and SD could be broadly grouped into two classes, those which altered their pattern of expression in response to the photoperiodic shift and those which altered the magnitude of their expression. Specifically, in Fig. 1, the majority of genes (six) change their pattern of expression over time between conditions (*MtFTb1*, *MtFTb2*, *MtPHYA*,
*MtGI*, *MtELF3a* and *MtSOC1a*), while two genes maintain a similar pattern, but vary in magnitude of expression between conditions (*MtFTa1* and the *NF-YC-like* gene). It was into these two classes that we decided to segment the time series data followed by clustering.

To classify the genes into these two classes we fit two models to the data. The first model included an interaction term between growth condition (SD or LD) and time of sampling (ZT0, ZT2 and ZT4). This was to identify genes which respond in a condition-specific manner over time such that any changes in the pattern of gene expression brought about by the photoperiodic shift are determined (e.g., *MtFTb1* in Fig. 1B). Secondly, we repeated the analysis with a model which lacked this interaction term to test for just the effect of condition (LD vs SD) on the magnitude of gene expression (e.g., the *NF-YC-like* gene in Fig. 1G). This dual approach facilitated the identification of genes which alter their pattern of expression or only the magnitude of their expression respectively.

Alongside this analysis, an *a priori* list of 146 candidate genes was also assembled consisting of genes shown to, or are suspected of, participating in flowering time regulation via the photoperiod pathway (Tables 1 and 2). In addition to the *FT-like* genes, the list incorporates photoreceptors, components of the circadian clock and classes of transcription factor which could potentially regulate the *FT-like* genes chosen based on their role in *A. thaliana*. These include candidates from the *CDF*, *TPL* and *NF-Y* families as well as additional transcription factors from families with members known to bind the promoter of *FT* in *A. thaliana* compiled by *Ridge et al. (2016)*. These include the genes containing CCT-domain and B-box domains as well as the *CIB/BEE-like* (*CBL*), *CRYPTOCHROME-INTERACTING BASIC-helix-loop-helix* (*CIB*), *PHYTOCHROME INTERACTING FACTOR* (*PIF*), and *APETALA2* (*AP2*) families.

### Photoperiod induced changes in the pattern of gene expression

To identify changes in the pattern of gene expression over time, a generalised linear model was fit using the growth condition and time of sampling as predictors along with an interaction term. A likelihood ratio test was then used to test for genes where this interaction term is significant relative to a reduced model lacking the interaction term. A significant result indicates that the expression of the gene responds in a condition-specific manner over time (i.e., a low *p*-value indicates a change in expression *pattern* not solely magnitude). This is illustrated in Fig. 1 where *MtPHYA*, *MtELF3a*, *MtGI*, *MtFTb1* and *MtSOC1a* have distinctly different patterns of expression over time in LD compared to SD. On the other hand *MtFTa1* and the *NF-YC-like* gene *Medtr1g082660* show a similar pattern in both LD and SD but at differing magnitudes so in this context are not considered to alter their pattern of expression. While *MtFTb2* also appears to alter it's pattern over time, it was not significant in the likelihood ratio test.

This approach resulted in 9,516 genes with altered expression ($\alpha = 0.05$; full results in Table S4) or 30.34% of those tested with >1 read. To aid interpretation of these changes and quickly identify the timepoint(s) at which individual genes differed in LD relative to SD, Wald significance tests were used to contrast the difference in expression between the two conditions at each of the three timepoints. The significance levels of these genes were adjusted for all three contrasts together using the false discovery rate method. This

Thomson et al. (2019), *PeerJ*, DOI 10.7717/peerj.6626

**Table 1** Photoperiod induced changes in the pattern of gene expression over time in candidate flowering time loci.

| Gene identifier | Name | Description (Mt4.0 annotation) | Interaction adj. $p$-value | ZT0 Mean read counts | ZT0 Log$_2$ fold-change | ZT0 Log$_2$ s.e | ZT0 adj. $p$-value | ZT2 Mean read counts | ZT2 Log$_2$ fold-change | ZT2 Log$_2$ s.e | ZT2 adj. $p$-value | ZT4 Meam read counts | ZT4 Log$_2$ fold-change | ZT4 Log$_2$ s.e | ZT4 adj. $p$-value | Cluster | Membership |
|---|---|---|---|---|---|---|---|---|---|---|---|---|---|---|---|---|---|
| *Medtr1g085160* | *MtPHYA* | phytochrome protein A | $3.20 \times 10^{-24}$ | 1100 | −0.13 | 0.10 | $2.30 \times 10^{-01}$ | 1000 | −0.71 | 0.10 | $5.80 \times 10^{-12}$ | 1900 | −1.60 | 0.10 | $6.90 \times 10^{-60}$ | 11 | 0.41 |
| *Medtr2g034040* | *MtPHYB* | phytochrome protein B | $2.70 \times 10^{-08}$ | 790 | −0.38 | 0.08 | $7.60 \times 10^{-06}$ | 800 | 0.06 | 0.08 | $5.50 \times 10^{-01}$ | 590 | 0.34 | 0.08 | $1.50 \times 10^{-04}$ | 18 | 0.74 |
| *Medtr2g049520* | *MtPHYE* | phytochrome protein | – | – | – | – | – | – | – | – | – | – | – | – | – | – | – |
| *Medtr5g063920* | *MtCRY1* | cryptochrome protein | $1.70 \times 10^{-13}$ | 5500 | −0.85 | 0.07 | $1.40 \times 10^{-29}$ | 3600 | −0.04 | 0.07 | $6.30 \times 10^{-01}$ | 4700 | −0.26 | 0.07 | $1.00 \times 10^{-03}$ | 10 | 0.53 |
| *Medtr1g076190* | *MtCrRY2A* | cryptochrome 2B apoprotein | $3.60 \times 10^{-21}$ | 2500 | −0.68 | 0.08 | $1.40 \times 10^{-16}$ | 1200 | 0.35 | 0.08 | $8.50 \times 10^{-05}$ | 2000 | −0.65 | 0.08 | $6.30 \times 10^{-15}$ | 15 | 0.47 |
| *Medtr1g043190* | *MtCRY2B* | cryptochrome 2B apoprotein | $4.60 \times 10^{-21}$ | 300 | −0.96 | 0.14 | $8.50 \times 10^{-12}$ | 140 | −1.60 | 0.16 | $7.50 \times 10^{-22}$ | 250 | −3.10 | 0.17 | $1.30 \times 10^{-73}$ | 10 | 0.18 |
| *Medtr7g118330* | *MtLHY* | late elongated hypocotyl-like protein | $3.00 \times 10^{-70}$ | 37000 | −1.90 | 0.18 | $2.00 \times 10^{-25}$ | 28000 | 0.57 | 0.18 | $2.60 \times 10^{-03}$ | 18000 | 2.90 | 0.18 | $7.40 \times 10^{-58}$ | 18 | 0.27 |
| *Medtr4g108880* | *MtTOC1a* | two-component response regulator-like APRR7 protein | $1.50 \times 10^{-58}$ | 75 | 4.40 | 0.36 | $2.10 \times 10^{-31}$ | 63 | 2.00 | 0.23 | $5.10 \times 10^{-17}$ | 120 | −0.97 | 0.18 | $2.60 \times 10^{-07}$ | 2 | 0.69 |
| *Medtr3g037390* | *MtTOC1b* | timing of cab expression 1/PRR response regulator | $2.00 \times 10^{-22}$ | 100 | 1.70 | 0.19 | $4.10 \times 10^{-18}$ | 100 | 0.88 | 0.18 | $2.30 \times 10^{-06}$ | 190 | −0.71 | 0.15 | $1.70 \times 10^{-05}$ | 2 | 0.37 |
| *Medtr4g061360* | *MtPRR37a* | PRR response regulator | $1.20 \times 10^{-11}$ | 580 | −4.40 | 0.21 | $1.80 \times 10^{-97}$ | 1600 | −3.40 | 0.18 | $5.30 \times 10^{-77}$ | 4400 | −2.40 | 0.17 | $1.10 \times 10^{-40}$ | 12 | 0.41 |
| *Medtr1g067110* | *MtPRR37b* | two-component response regulator-like APRR7 protein | $1.90 \times 10^{-03}$ | 350 | −2.00 | 0.14 | $3.30 \times 10^{-46}$ | 820 | −1.40 | 0.12 | $5.20 \times 10^{-30}$ | 2700 | −2.10 | 0.12 | $5.60 \times 10^{-66}$ | 9 | 0.34 |
| *Medtr3g092780* | *MtPRR59a* | PRR response regulator | $4.20 \times 10^{-30}$ | 22 | 0.28 | 0.32 | $4.60 \times 10^{-01}$ | 150 | −2.30 | 0.24 | $4.40 \times 10^{-20}$ | 1400 | −4.30 | 0.21 | $4.20 \times 10^{-88}$ | 9 | 0.5 |
| *Medtr8g024260* | *MtPRR59b* | PRR response regulator | $1.40 \times 10^{-05}$ | 54 | −2.30 | 0.24 | $5.20 \times 10^{-21}$ | 210 | −1.20 | 0.15 | $1.50 \times 10^{-14}$ | 1300 | −2.00 | 0.13 | $5.50 \times 10^{-56}$ | 9 | 0.33 |
| *Medtr7g118260* | *MtPRR59c* | PRR response regulator | $9.50 \times 10^{-34}$ | 200 | −6.20 | 0.38 | $8.60 \times 10^{-57}$ | 710 | −2.00 | 0.18 | $3.30 \times 10^{-26}$ | 2100 | −1.90 | 0.18 | $4.30 \times 10^{-25}$ | 12 | 0.59 |
| *Medtr3g103970* | *MtELF3a* | early flowering protein | $1.30 \times 10^{-43}$ | 250 | 3.70 | 0.24 | $7.30 \times 10^{-50}$ | 43 | 2.00 | 0.30 | $1.60 \times 10^{-10}$ | 110 | −1.00 | 0.22 | $2.20 \times 10^{-05}$ | 4 | 0.37 |
| *Medtr1g016920* | *MtELF3b* | EARLY flowering protein, putative | $9.20 \times 10^{-01}$ | – | – | – | – | – | – | – | – | – | – | – | – | – | – |
| *Medtr8g015470* | *ELF3-like* | hypothetical protein | – | – | – | – | – | – | – | – | – | – | – | – | – | – | – |
| *Medtr8g015480* | *ELF3-like* | early flowering protein, putative | $4.50 \times 10^{-12}$ | 160 | 1.50 | 0.23 | $4.20 \times 10^{-10}$ | 56 | −0.38 | 0.26 | $2.00 \times 10^{-01}$ | 150 | −0.86 | 0.23 | $4.40 \times 10^{-04}$ | 16 | 0.34 |
| *Medtr4g064730* | *MtLUXa* | myb-like DNA-binding domain, shaqkyf class protein | $1.60 \times 10^{-07}$ | 11 | 3.80 | 0.79 | $6.10 \times 10^{-06}$ | 12 | 0.36 | 0.45 | $5.00 \times 10^{-01}$ | 21 | −0.62 | 0.38 | $1.50 \times 10^{-01}$ | 7 | 0.57 |
| *Medtr7g089010* | *MtLUXb* | MYB-like transcription factor family protein | $2.90 \times 10^{-02}$ | 180 | −1.60 | 0.14 | $6.60 \times 10^{-28}$ | 310 | −1.10 | 0.12 | $4.90 \times 10^{-19}$ | 330 | −1.10 | 0.12 | $7.70 \times 10^{-17}$ | 12 | 0.37 |
| *Medtr3g070490* | *MtELF4* | early flowering protein | $2.10 \times 10^{-71}$ | 53 | 3.60 | 0.35 | $2.40 \times 10^{-22}$ | 24 | 1.80 | 0.33 | $4.20 \times 10^{-07}$ | 56 | −4.00 | 0.37 | $2.30 \times 10^{-25}$ | 16 | 0.32 |
| *Medtr8g020200* | *ELF4-like* | early flowering protein | $5.80 \times 10^{-01}$ | – | – | – | – | – | – | – | – | – | – | – | – | – | – |
| *Medtr4g125590* | *ELF4-like* | early flowering protein | $9.20 \times 10^{-02}$ | – | – | – | – | – | – | – | – | – | – | – | – | – | – |
| *Medtr2g041310* | *ELF4-like* | early flowering protein | $2.60 \times 10^{-02}$ | 160 | −1.40 | 0.20 | $8.50 \times 10^{-11}$ | 120 | −0.55 | 0.20 | $1.20 \times 10^{-02}$ | 200 | −0.67 | 0.19 | $1.10 \times 10^{-03}$ | 10 | 0.17 |
| *Medtr1g098160* | *MtGI* | gigantea protein 1B | $2.60 \times 10^{-14}$ | 35 | −3.40 | 0.36 | $5.90 \times 10^{-20}$ | 3300 | −0.86 | 0.12 | $8.40 \times 10^{-12}$ | 4100 | −1.70 | 0.12 | $1.20 \times 10^{-45}$ | 9 | 0.23 |
| *Medtr8g105590* | *MtFKF1* | flavin-binding kelch repeat F-box protein, putative | $4.60 \times 10^{-56}$ | 14 | 3.30 | 0.60 | $1.50 \times 10^{-07}$ | 24 | −1.40 | 0.33 | $5.40 \times 10^{-05}$ | 250 | −5.00 | 0.29 | $1.70 \times 10^{-64}$ | 11 | 0.78 |

**Table 1** (*continued*)

| Gene identifier | Name | Description (Mt4.0 annotation) | Interaction adj. *p*-value | ZT0 | | | | ZT2 | | | | ZT4 | | | | Cluster | Membership |
|---|---|---|---|---|---|---|---|---|---|---|---|---|---|---|---|---|---|
| | | | | Mean read counts | Log$_2$ fold-change | Log$_2$ s.e | adj. *p*-value | Mean read counts | Log$_2$ fold-change | Log$_2$ s.e | adj. *p*-value | Meam read counts | Log$_2$ fold-change | Log$_2$ s.e | adj. *p*-value | | |
| *Medtr2g036510* | *MtZTL* | galactose oxidase/kelch repeat protein | $6.90 \times 10^{-01}$ | – | – | – | – | – | – | – | – | – | – | – | – | – | – |
| *Medtr2g058520* | *MtE1* | E1 protein | $2.20 \times 10^{-03}$ | 1.8 | −2.30 | 1.20 | $8.60 \times 10^{-02}$ | **23** | **2.20** | **0.51** | **$6.20 \times 10^{-05}$** | **23** | **1.40** | **0.49** | **$7.10 \times 10^{-03}$** | 8 | 0.16 |
| *Medtr7g084970* | *MtFTa1* | flowering locus protein T | $5.70 \times 10^{-01}$ | – | – | – | – | – | – | – | – | – | – | – | – | – | – |
| *Medtr7g085020* | *MtFTa2* | flowering locus protein T | $2.80 \times 10^{-03}$ | **17** | **1.00** | **0.44** | **$3.80 \times 10^{-02}$** | **11** | **−1.10** | **0.49** | **$4.90 \times 10^{-02}$** | 28 | −0.97 | 0.39 | **$2.30 \times 10^{-02}$** | 11 | 0.3 |
| *Medtr6g033040* | *MtFTa3* | flowering locus protein T | – | – | – | – | – | – | – | – | – | – | – | – | – | – | – |
| *Medtr7g006630* | *MtFTb1* | flowering locus protein T | $2.00 \times 10^{-03}$ | 0 | −0.16 | 1.70 | $9.40 \times 10^{-01}$ | 3 | 5.00 | 1.30 | **$2.10 \times 10^{-04}$** | **90** | **9.90** | **1.20** | **$2.60 \times 10^{-15}$** | 3 | 0.4 |
| *Medtr7g006690* | *MtFTb2* | flowering locus protein T | $6.70 \times 10^{-01}$ | – | – | – | – | – | – | – | – | | | | | | |
| *Medtr2g461760* | *MtFULa* | MADS-box transcription factor | – | – | – | – | – | – | – | – | – | – | – | – | – | – | – |
| *Medtr4g109830* | *MtFULb* | MADS-box transcription factor | $3.00 \times 10^{-02}$ | 10 | 1.70 | 0.60 | **$8.00 \times 10^{-03}$** | 24 | −0.27 | 0.46 | $6.30 \times 10^{-01}$ | **15** | **1.40** | **0.52** | **$1.60 \times 10^{-02}$** | 8 | 0.39 |
| *Medtr7g016630* | *MtFULc* | MADS-box transcription factor | – | – | – | – | – | – | – | – | – | – | – | – | – | – | – |
| *Medtr7g075870* | *MtSOC1a* | MADS-box transcription factor | $7.40 \times 10^{-04}$ | 110 | −0.16 | 0.18 | $4.60 \times 10^{-01}$ | 120 | 0.53 | 0.18 | **$5.30 \times 10^{-03}$** | 150 | 0.86 | 0.17 | **$1.90 \times 10^{-06}$** | 3 | 0.35 |
| *Medtr8g033250* | *MtSOC1b* | MADS-box transcription factor | – | – | – | – | – | – | – | – | – | – | – | – | – | – | – |
| *Medtr8g033220* | *MtSOC1c* | MADS-box transcription factor | $9.80 \times 10^{-01}$ | – | – | – | – | – | – | – | – | – | – | – | – | – | – |
| *Medtr2g059540* | *MtCDF1* | Dof domain zinc finger protein | $8.80 \times 10^{-03}$ | 0 | −0.16 | 1.70 | $9.40 \times 10^{-01}$ | **8.7** | **−4.70** | **1.10** | **$4.30 \times 10^{-05}$** | **96** | **−8.20** | **1.10** | **$1.00 \times 10^{-13}$** | 11 | 0.4 |
| *Medtr5g041420* | *MtCDF2* | DOF zinc finger protein | $1.40 \times 10^{-05}$ | 11 | −1.10 | 0.54 | $5.70 \times 10^{-02}$ | 9.5 | 2.40 | 0.65 | **$7.60 \times 10^{-04}$** | 2.7 | 3.80 | 1.30 | **$7.30 \times 10^{-03}$** | 5 | 0.19 |
| *Medtr5g041530* | *MtCDF3* | cycling DOF factor 2 | $3.10 \times 10^{-19}$ | 220 | 0.29 | 0.16 | $9.30 \times 10^{-02}$ | **170** | **1.30** | **0.17** | **$8.80 \times 10^{-14}$** | 99 | 2.80 | 0.23 | **$1.20 \times 10^{-34}$** | 1 | 0.58 |
| *Medtr6g012450* | *MtCDF4* | DOF-type zinc finger DNA-binding family protein | $3.40 \times 10^{-77}$ | **860** | **−1.40** | **0.14** | **$1.60 \times 10^{-21}$** | **350** | **1.90** | **0.16** | **$5.40 \times 10^{-31}$** | **270** | **2.40** | **0.17** | **$1.30 \times 10^{-45}$** | 5 | 0.34 |
| *Medtr6g027460* | *MtCDF5* | Dof zinc finger DOF5.2-like protein | $1.40 \times 10^{-18}$ | **370** | **1.40** | **0.19** | **$4.40 \times 10^{-13}$** | **350** | **2.00** | **0.19** | **$2.30 \times 10^{-24}$** | **100** | **4.90** | **0.37** | **$1.30 \times 10^{-37}$** | 1 | 0.69 |
| *Medtr7g010950* | *MtCDF6* | DOF-type zinc finger DNA-binding family protein | $6.40 \times 10^{-30}$ | 360 | −0.40 | 0.16 | **$2.50 \times 10^{-02}$** | **260** | **1.60** | **0.17** | **$1.10 \times 10^{-19}$** | **200** | **2.40** | **0.19** | **$2.40 \times 10^{-36}$** | 1 | 0.32 |
| *Medtr2g016030* | *MtCDFa* | Dof domain zinc finger protein | $8.90 \times 10^{-01}$ | – | – | – | – | – | – | – | – | – | – | – | – | – | – |
| *Medtr3g435480* | *MtCDFb* | DOF-type zinc finger DNA-binding family protein | $2.60 \times 10^{-22}$ | **1400** | **−3.10** | **0.24** | **$2.40 \times 10^{-35}$** | 800 | −0.33 | 0.24 | $2.30 \times 10^{-01}$ | 1400 | 0.33 | 0.24 | $2.20 \times 10^{-01}$ | 12 | 0.23 |
| *Medtr4g082060* | *MtCDFc* | DOF-type zinc finger DNA-binding family protein | $1.90 \times 10^{-41}$ | **970** | **−1.90** | **0.15** | **$5.40 \times 10^{-35}$** | 720 | 0.73 | 0.15 | **$2.80 \times 10^{-06}$** | 760 | 0.69 | 0.15 | **$1.10 \times 10^{-05}$** | 13 | 0.26 |
| *Medtr5g041380* | *MtCDFd* | DOF domain, zinc finger protein | $6.00 \times 10^{-01}$ | – | – | – | – | – | – | – | – | – | – | – | – | – | – |
| *Medtr5g041400* | *MtCDFe* | DOF domain, zinc finger protein | $2.70 \times 10^{-06}$ | 8.3 | −2.70 | 0.71 | **$2.80 \times 10^{-04}$** | 8.7 | 1.00 | 0.58 | $1.20 \times 10^{-01}$ | 3.5 | 2.50 | 0.98 | **$1.70 \times 10^{-02}$** | 18 | 0.21 |
| *Medtr6g027450* | *MtCDFf* | Dof zinc finger DOF5.2-like protein | $3.30 \times 10^{-01}$ | – | – | – | – | – | – | – | – | – | – | – | – | – | – |

**Table 1** (*continued*)

| Gene identifier | Name | Description (Mt4.0 annotation) | Interaction adj. *p*-value | ZT0 | | | | ZT2 | | | | ZT4 | | | | Cluster | Membership |
|---|---|---|---|---|---|---|---|---|---|---|---|---|---|---|---|---|---|
| | | | | Mean read counts | Log$_2$ fold-change | Log$_2$ s.e | adj. *p*-value | Mean read counts | Log$_2$ fold-change | Log$_2$ s.e | adj. *p*-value | Meam read counts | Log$_2$ fold-change | Log$_2$ s.e | adj. *p*-value | | |
| *Medtr7g086780* | *MtCDFg* | Dof zinc finger DOF5.2-like protein | $1.80 \times 10^{-01}$ | – | – | – | – | – | – | – | – | – | – | – | – | – | – |
| *Medtr8g044220* | *MtCDFh* | DOF-type zinc finger DNA-binding family protein | $1.50 \times 10^{-01}$ | – | – | – | – | – | – | – | – | – | – | – | – | – | – |
| *Medtr5g085250* | *MtCOP1* | E3 ubiquitin-protein ligase COP1 | **$1.70 \times 10^{-02}$** | 1200 | −0.48 | 0.09 | **$3.20 \times 10^{-07}$** | 1800 | −0.26 | 0.09 | **$6.10 \times 10^{-03}$** | 1600 | −0.08 | 0.09 | $4.50 \times 10^{-01}$ | 8 | 0.17 |
| *Medtr5g009530* | *SPA1-like* | ubiquitin ligase cop1, putative | $1.90 \times 10^{-01}$ | – | – | – | – | – | – | – | – | – | – | – | – | – | – |
| *Medtr8g027985* | *SPA1-like* | ubiquitin ligase cop1, putative | **$8.90 \times 10^{-170}$** | **310** | **3.40** | **0.16** | **$5.10 \times 10^{-96}$** | **1000** | **−1.10** | **0.11** | **$4.10 \times 10^{-20}$** | **2400** | **−1.60** | **0.11** | **$1.60 \times 10^{-47}$** | 17 | 0.46 |
| *Medtr2g085210* | *SPA1-like* | ubiquitin ligase cop1, putative | $1.30 \times 10^{-01}$ | – | – | – | – | – | – | – | – | – | – | – | – | – | – |
| *Medtr2g104140* | *TPL-like* | topless-like protein | **$9.70 \times 10^{-05}$** | 1700 | −0.89 | 0.12 | **$3.30 \times 10^{-13}$** | 1800 | −0.48 | 0.12 | **$1.50 \times 10^{-04}$** | 1500 | −0.11 | 0.12 | $4.20 \times 10^{-01}$ | 18 | 0.2 |
| *Medtr4g009840* | *TPL-like* | topless-like protein | $6.50 \times 10^{-01}$ | – | – | – | – | – | – | – | – | – | – | – | – | – | – |
| *Medtr4g114980* | *TPL-like* | topless-like protein | $3.50 \times 10^{-01}$ | – | – | – | – | – | – | – | – | – | – | – | – | – | – |
| *Medtr2g435370* | *TPL-like* | transducin family protein/WD-40 repeat protein | – | – | – | – | – | – | – | – | – | – | – | – | – | – | – |
| *Medtr2g435440* | *TPL-like* | topless-like protein | – | – | – | – | – | – | – | – | – | – | – | – | – | – | – |
| *Medtr4g120900* | *TPL-like* | topless-like protein | **$1.10 \times 10^{-04}$** | 980 | −0.48 | 0.07 | **$9.70 \times 10^{-12}$** | 1200 | −0.22 | 0.07 | **$1.70 \times 10^{-03}$** | 1400 | −0.05 | 0.06 | $5.40 \times 10^{-01}$ | 12 | 0.34 |
| *Medtr7g112460* | *TPL-like* | topless-like protein | $7.60 \times 10^{-01}$ | – | – | – | – | – | – | – | – | – | – | – | – | – | – |
| *Medtr2g065670* | *TPL-like* | topless-like protein | $8.50 \times 10^{-01}$ | – | – | – | – | – | – | – | – | – | – | – | – | – | – |
| *Medtr2g435380* | *TPL-like* | topless-like protein | – | – | – | – | – | – | – | – | – | – | – | – | – | – | – |
| *Medtr1g012820* | *TPL-like* | topless-like protein | – | – | – | – | – | – | – | – | – | – | – | – | – | – | – |
| *Medtr6g444980* | *MtFE* | myb-like transcription factor family protein | **$3.00 \times 10^{-04}$** | 430 | 0.46 | 0.10 | **$6.50 \times 10^{-06}$** | 380 | 0.03 | 0.10 | $7.90 \times 10^{-01}$ | 440 | 0.61 | 0.10 | **$8.50 \times 10^{-10}$** | 1 | 0.29 |
| *Medtr3g058980* | *NF-YB like* | nuclear transcription factor Y protein | **$3.60 \times 10^{-04}$** | 910 | 0.09 | 0.09 | $3.80 \times 10^{-01}$ | 540 | −0.42 | 0.09 | **$1.30 \times 10^{-05}$** | 820 | −0.31 | 0.09 | **$8.40 \times 10^{-04}$** | 15 | 0.21 |
| *Medtr5g095740* | *NF-YB like* | nuclear transcription factor Y protein | **$2.40 \times 10^{-09}$** | 570 | −0.80 | 0.17 | **$4.70 \times 10^{-06}$** | 160 | 0.69 | 0.18 | **$5.00 \times 10^{-04}$** | 430 | −0.74 | 0.17 | **$3.30 \times 10^{-05}$** | 15 | 0.54 |
| *Medtr1g082660* | *NF-YC like* | nuclear transcription factor Y protein | $8.80 \times 10^{-02}$ | – | – | – | – | – | – | – | – | – | – | – | – | – | – |
| *Medtr3g099180* | *NF-YC like* | nuclear transcription factor Y protein | **$5.10 \times 10^{-03}$** | 1800 | 0.51 | 0.12 | **$5.40 \times 10^{-05}$** | 1200 | 0.84 | 0.12 | **$3.20 \times 10^{-11}$** | 1900 | 0.22 | 0.12 | $9.40 \times 10^{-02}$ | 4 | 0.23 |
| *Medtr7g113680* | *NF-YC like* | nuclear transcription factor Y protein | $9.00 \times 10^{-01}$ | – | – | – | – | – | – | – | – | – | – | – | – | – | – |
| *Medtr1g093600* | *MtTEM1* | AP2/ERF and B3 domain transcription factor | **$3.00 \times 10^{-09}$** | **2600** | **−1.50** | **0.16** | **$3.90 \times 10^{-20}$** | 560 | −0.15 | 0.17 | $4.50 \times 10^{-01}$ | **360** | **−1.50** | **0.17** | **$1.30 \times 10^{-16}$** | 14 | 0.46 |
| *Medtr5g053920* | *MtTEM2* | AP2/ERF and B3 domain transcription factor | **$7.40 \times 10^{-03}$** | **1900** | **−1.20** | **0.17** | **$2.10 \times 10^{-11}$** | 1000 | −0.35 | 0.18 | $7.60 \times 10^{-02}$ | 550 | −0.79 | 0.18 | **$3.30 \times 10^{-05}$** | 14 | 0.23 |
| *Medtr4g061200* | *MtTOE1a* | AP2-like ethylene-responsive transcription factor | **$1.90 \times 10^{-05}$** | **42** | **1.10** | **0.28** | **$1.90 \times 10^{-04}$** | 37 | 0.61 | 0.28 | **$4.80 \times 10^{-02}$** | 70 | −0.67 | 0.25 | **$1.30 \times 10^{-02}$** | 2 | 0.25 |
| *Medtr2g093060* | *MtTOE1b* | AP2-like ethylene-responsive transcription factor | **$1.20 \times 10^{-04}$** | 1200 | −0.15 | 0.08 | $1.00 \times 10^{-01}$ | 1100 | −0.64 | 0.08 | **$2.20 \times 10^{-14}$** | 1600 | −0.24 | 0.08 | **$4.90 \times 10^{-03}$** | 9 | 0.2 |

| Gene identifier | Name | Description (Mt4.0 annotation) | Interaction adj. $p$-value | ZT0 | | | | ZT2 | | | | ZT4 | | | | Cluster | Membership |
|---|---|---|---|---|---|---|---|---|---|---|---|---|---|---|---|---|---|
| | | | | Mean read counts | Log$_2$ fold-change | Log$_2$ s.e | adj. $p$-value | Mean read counts | Log$_2$ fold-change | Log$_2$ s.e | adj. $p$-value | Meam read counts | Log$_2$ fold-change | Log$_2$ s.e | adj. $p$-value | | |
| *Medtr7g100590* | *MtTOE1c* | AP2 domain transcription factor | **1.90 $\times 10^{-02}$** | 49 | 0.82 | 0.27 | **6.00 $\times 10^{-03}$** | 28 | 0.30 | 0.30 | 4.00 $\times 10^{-01}$ | 64 | −0.36 | 0.26 | 2.10 $\times 10^{-01}$ | 16 | 0.38 |
| *Medtr1g049140* | *MtTOE2* | AP2 domain transcription factor | **2.50 $\times 10^{-17}$** | **1600** | **−1.60** | **0.13** | **4.00 $\times 10^{-33}$** | 2500 | −0.65 | 0.13 | **8.40 $\times 10^{-07}$** | 2900 | 0.06 | 0.13 | 6.80 $\times 10^{-01}$ | 12 | 0.71 |
| *Medtr5g016810* | *MtAP2a* | AP2 domain transcription factor | **2.30 $\times 10^{-14}$** | 510 | 0.57 | 0.11 | **2.80 $\times 10^{-07}$** | 880 | −0.61 | 0.10 | **4.30 $\times 10^{-09}$** | 730 | −0.24 | 0.10 | **3.30 $\times 10^{-02}$** | 6 | 0.46 |
| *Medtr4g094868* | *MtAP2b* | AP2 domain transcription factor | **1.50 $\times 10^{-12}$** | **260** | **1.00** | **0.13** | **7.80 $\times 10^{-14}$** | 390 | −0.17 | 0.12 | 2.30 $\times 10^{-01}$ | 280 | −0.24 | 0.13 | 9.70 $\times 10^{-02}$ | 6 | 0.21 |
| *Medtr7g018170* | *MtCOLa* | zinc finger constans-like protein | **5.90 $\times 10^{-104}$** | **4400** | **−2.30** | **0.11** | **3.60 $\times 10^{-105}$** | 1500 | 0.79 | 0.11 | **1.50 $\times 10^{-12}$** | 2200 | 0.38 | 0.11 | **8.10 $\times 10^{-04}$** | 13 | 0.36 |
| *Medtr1g013450* | *MtCOLb* | zinc finger constans-like protein | **1.60 $\times 10^{-47}$** | 5000 | −0.82 | 0.07 | **8.30 $\times 10^{-27}$** | 3800 | 0.16 | 0.08 | 5.50 $\times 10^{-02}$ | 3100 | 0.76 | 0.08 | **1.20 $\times 10^{-22}$** | 18 | 0.37 |
| *Medtr3g105710* | *MtCOLc* | zinc finger constans-like protein | **1.30 $\times 10^{-28}$** | 1700 | −0.48 | 0.05 | **8.30 $\times 10^{-22}$** | 860 | 0.19 | 0.06 | **1.40 $\times 10^{-03}$** | 1200 | 0.28 | 0.05 | **5.00 $\times 10^{-07}$** | 14 | 0.32 |
| *Medtr4g128930* | *MtCOLd* | zinc finger constans-like protein | **1.10 $\times 10^{-38}$** | **2400** | **−2.60** | **0.14** | **6.10 $\times 10^{-81}$** | 1500 | −0.34 | 0.14 | **2.00 $\times 10^{-02}$** | 2300 | −0.35 | 0.13 | **1.50 $\times 10^{-02}$** | 12 | 0.19 |
| *Medtr3g082630* | *MtCOLe* | B-box type zinc finger protein | **1.50 $\times 10^{-54}$** | **48** | **7.90** | **1.20** | **3.80 $\times 10^{-10}$** | 5 | 2.30 | 0.75 | **5.20 $\times 10^{-03}$** | **39** | **−3.90** | **0.46** | **4.60 $\times 10^{-16}$** | 16 | 0.47 |
| *Medtr5g069480* | *MtCOLf* | zinc finger constans-like protein | **9.40 $\times 10^{-12}$** | **490** | **4.20** | **0.21** | **1.40 $\times 10^{-88}$** | **130** | **2.20** | **0.21** | **4.50 $\times 10^{-24}$** | **110** | **2.40** | **0.23** | **4.10 $\times 10^{-25}$** | 4 | 0.57 |
| *Medtr7g108150* | *MtCOLg* | zinc finger constans-like protein | **6.10 $\times 10^{-03}$** | 880 | −0.36 | 0.09 | **7.50 $\times 10^{-05}$** | 510 | 0.08 | 0.09 | 4.40 $\times 10^{-01}$ | 650 | −0.20 | 0.09 | **3.60 $\times 10^{-02}$** | 14 | 0.38 |
| *Medtr7g083540* | *MtCOLh* | zinc finger constans-like protein | **6.20 $\times 10^{-07}$** | 57 | 0.14 | 0.24 | 6.30 $\times 10^{-01}$ | 69 | −0.19 | 0.23 | 5.00 $\times 10^{-01}$ | **61** | **−1.70** | **0.26** | **8.40 $\times 10^{-11}$** | 16 | 0.1 |
| *Medtr8g104190* | *MtCOLi* | zinc finger constans-like protein | 5.50 $\times 10^{-02}$ | – | – | – | – | – | – | – | – | – | – | – | – | – | – |
| *Medtr2g088900* | *MtCOLj* | zinc finger constans-like protein | **2.50 $\times 10^{-02}$** | 79 | −0.81 | 0.29 | **1.10 $\times 10^{-02}$** | 83 | 0.28 | 0.29 | 4.00 $\times 10^{-01}$ | 75 | 0.30 | 0.29 | 3.80 $\times 10^{-01}$ | 18 | 0.28 |
| *Medtr1g110870* | *MtCOLk* | zinc finger constans-like protein | **1.40 $\times 10^{-19}$** | **2000** | **1.70** | **0.16** | **2.20 $\times 10^{-22}$** | 230 | −0.05 | 0.18 | 8.20 $\times 10^{-01}$ | 1200 | −0.53 | 0.17 | **3.10 $\times 10^{-03}$** | 15 | 0.19 |
| *Medtr7g032240* | *MtCMF1* | CCT motif protein | **7.90 $\times 10^{-04}$** | 910 | −0.26 | 0.08 | **2.10 $\times 10^{-03}$** | 470 | 0.11 | 0.09 | 2.50 $\times 10^{-01}$ | 1100 | 0.16 | 0.08 | 5.40 $\times 10^{-02}$ | 15 | 0.34 |
| *Medtr4g127420* | *MtCMF2* | import apparatus protein | **2.40 $\times 10^{-19}$** | 180 | 0.35 | 0.13 | **1.60 $\times 10^{-02}$** | 660 | −0.78 | 0.11 | **6.00 $\times 10^{-12}$** | **840** | **−1.30** | **0.11** | **3.50 $\times 10^{-31}$** | 9 | 0.43 |
| *Medtr5g072780* | *MtCMF3* | CCT motif protein | 8.60 $\times 10^{-02}$ | – | – | – | – | – | – | – | – | – | – | – | – | – | – |
| *Medtr3g100040* | *MtCMF5* | GATA transcription factor | 3.00 $\times 10^{-01}$ | – | – | – | – | – | – | – | – | – | – | – | – | – | – |
| *Medtr3g100050* | *MtCMF6* | GATA transcription factor | 6.80 $\times 10^{-01}$ | – | – | – | – | – | – | – | – | – | – | – | – | – | – |
| *Medtr5g066510* | *MtCMF7* | GATA transcription factor | 7.30 $\times 10^{-01}$ | – | – | – | – | – | – | – | – | – | – | – | – | – | – |
| *Medtr4g093730* | *MtCMF8* | GATA transcription factor | 4.30 $\times 10^{-01}$ | – | – | – | – | – | – | – | – | – | – | – | – | – | – |
| *Medtr1g008220* | *MtCMF9* | CCT motif protein | **3.40 $\times 10^{-21}$** | **480** | **1.20** | **0.18** | **6.30 $\times 10^{-11}$** | **500** | **−1.30** | **0.18** | **2.00 $\times 10^{-12}$** | 50 | −0.54 | 0.24 | **3.90 $\times 10^{-02}$** | 18 | 0.27 |
| *Medtr3g091340* | *MtCMF10* | CCT motif protein | **7.60 $\times 10^{-43}$** | **470** | **1.90** | **0.14** | **3.20 $\times 10^{-42}$** | 700 | −0.79 | 0.13 | **6.90 $\times 10^{-09}$** | 100 | 0.66 | 0.17 | **2.90 $\times 10^{-04}$** | 18 | 0.18 |
| *Medtr4g061910* | *MtCMF11a* | CCT motif protein | – | – | – | – | – | – | – | – | – | – | – | – | – | – | – |
| *Medtr4g061823* | *MtCMF11b* | CCT motif protein | – | – | – | – | – | – | – | – | – | – | – | – | – | – | – |
| *Medtr2g096080* | *MtCMF12* | CCT motif protein | 7.90 $\times 10^{-02}$ | – | – | – | – | – | – | – | – | – | – | – | – | – | – |

Peer J

**Table 1** (*continued*)

| Gene identifier | Name | Description (Mt4.0 annotation) | Interaction adj. *p*-value | ZT0 | | | | ZT2 | | | | ZT4 | | | | Cluster | Membership |
|---|---|---|---|---|---|---|---|---|---|---|---|---|---|---|---|---|---|
| | | | | Mean read counts | Log$_2$ fold-change | Log$_2$ s.e | adj. *p*-value | Mean read counts | Log$_2$ fold-change | Log$_2$ s.e | adj. *p*-value | Meam read counts | Log$_2$ fold-change | Log$_2$ s.e | adj. *p*-value | | |
| *Medtr8g098725* | *MtCMF13* | CCT motif protein | – | – | – | – | – | – | – | – | – | – | – | – | – | – | – |
| *Medtr5g010120* | *MtCMF14* | CCT motif protein | – | – | – | – | – | – | – | – | – | – | – | – | – | – | – |
| *Medtr2g068730* | *MtCMF15* | CCT motif protein | – | – | – | – | – | – | – | – | – | – | – | – | – | – | – |
| *Medtr1g073350* | *MtCMF16* | CCT motif protein | $4.50 \times 10^{-02}$ | 42 | 0.67 | 0.39 | $1.30 \times 10^{-01}$ | 16 | −0.75 | 0.46 | $1.50 \times 10^{-01}$ | 36 | −0.75 | 0.40 | $9.50 \times 10^{-02}$ | 16 | 0.21 |
| *Medtr1g044785* | *MtCMF17* | CCT motif protein | $4.50 \times 10^{-01}$ | – | – | – | – | – | – | – | – | – | – | – | – | – | – |
| *Medtr4g008090* | *MtCMF18* | GATA transcription factor, putative | – | – | – | – | – | – | – | – | – | – | – | – | – | – | – |
| *Medtr1g023260* | CCT domain gene | salt tolerance-like protein | $1.20 \times 10^{-06}$ | 230 | 0.71 | 0.13 | $1.10 \times 10^{-07}$ | 200 | −0.28 | 0.13 | $5.30 \times 10^{-02}$ | 120 | 0.43 | 0.14 | $6.10 \times 10^{-03}$ | 18 | 0.2 |
| *Medtr1g109350* | CCT domain gene | B-box zinc finger protein, putative | $5.10 \times 10^{-05}$ | 990 | 1.30 | 0.19 | $6.60 \times 10^{-10}$ | 100 | 2.40 | 0.25 | $5.90 \times 10^{-21}$ | 240 | 0.92 | 0.21 | $2.80 \times 10^{-05}$ | 15 | 0.28 |
| *Medtr2g011450* | CCT domain gene | B-box type zinc finger protein | $1.50 \times 10^{-05}$ | 11 | −1.70 | 0.52 | $2.30 \times 10^{-03}$ | 30 | 0.32 | 0.38 | $4.70 \times 10^{-01}$ | 10 | 1.90 | 0.54 | $8.60 \times 10^{-04}$ | 8 | 0.37 |
| *Medtr2g073370* | CCT domain gene | B-box type zinc finger protein | $2.70 \times 10^{-15}$ | 1.8 | 4.20 | 1.30 | $3.00 \times 10^{-03}$ | 24 | −0.69 | 0.42 | $1.40 \times 10^{-01}$ | 66 | −4.90 | 0.53 | $4.00 \times 10^{-19}$ | 9 | 0.4 |
| *Medtr2g089310* | CCT domain gene | B-box type zinc finger protein | $2.50 \times 10^{-62}$ | 690 | −2.90 | 0.17 | $2.10 \times 10^{-67}$ | 750 | 0.13 | 0.16 | $5.00 \times 10^{-01}$ | 590 | 0.92 | 0.16 | $4.10 \times 10^{-08}$ | 12 | 0.18 |
| *Medtr2g099010* | CCT domain gene | salt tolerance-like protein | $1.00 \times 10^{-42}$ | 18000 | −0.59 | 0.10 | $3.40 \times 10^{-08}$ | 14000 | 0.25 | 0.10 | $2.30 \times 10^{-02}$ | 12000 | 1.50 | 0.10 | $2.90 \times 10^{-45}$ | 1 | 0.24 |
| *Medtr3g113070* | CCT domain gene | salt tolerance-like protein | – | – | – | – | – | – | – | – | – | – | – | – | – | – | – |
| *Medtr3g117320* | CCT domain gene | salt tolerance-like protein | $2.10 \times 10^{-04}$ | 26 | 3.50 | 0.64 | $2.70 \times 10^{-07}$ | 3.2 | 1.90 | 0.94 | $7.20 \times 10^{-02}$ | 8.3 | −0.48 | 0.62 | $5.20 \times 10^{-01}$ | 4 | 0.46 |
| *Medtr4g008050* | CCT domain gene | B-box type zinc finger protein, putative | $2.30 \times 10^{-11}$ | 49 | −0.70 | 0.27 | $1.70 \times 10^{-02}$ | 530 | −0.09 | 0.21 | $7.40 \times 10^{-01}$ | 350 | 1.60 | 0.22 | $3.90 \times 10^{-13}$ | 8 | 0.26 |
| *Medtr4g046640* | CCT domain gene | B-box type zinc finger protein | $3.70 \times 10^{-08}$ | 12 | 0.42 | 0.53 | $5.10 \times 10^{-01}$ | 81 | 1.20 | 0.42 | $7.60 \times 10^{-03}$ | 34 | 5.30 | 0.76 | $1.70 \times 10^{-11}$ | 8 | 0.65 |
| *Medtr4g067320* | CCT domain gene | salt tolerance-like protein | $6.60 \times 10^{-33}$ | 1200 | −2.10 | 0.10 | $9.10 \times 10^{-91}$ | 1800 | −0.45 | 0.10 | $2.00 \times 10^{-05}$ | 3700 | −0.71 | 0.10 | $2.60 \times 10^{-12}$ | 12 | 0.65 |
| *Medtr4g071200* | CCT domain gene | salt tolerance-like protein | – | – | – | – | – | – | – | – | – | – | – | – | – | – | – |
| *Medtr5g021580* | CCT domain gene | salt tolerance-like protein | $3.00 \times 10^{-08}$ | 1200 | −1.40 | 0.17 | $4.90 \times 10^{-15}$ | 1600 | −0.37 | 0.17 | $4.30 \times 10^{-02}$ | 1600 | 0.09 | 0.17 | $6.60 \times 10^{-01}$ | 12 | 0.48 |
| *Medtr3g116770* | *MtCBL1* | BHLH transcription factor | $5.20 \times 10^{-05}$ | 720 | −0.76 | 0.17 | $2.00 \times 10^{-05}$ | 320 | 0.29 | 0.17 | $1.30 \times 10^{-01}$ | 1300 | −0.68 | 0.16 | $1.20 \times 10^{-04}$ | 10 | 0.16 |

## Table 1 (continued)

| Gene identifier | Name | Description (Mt4.0 annotation) | Interaction adj. p-value | ZT0 Mean read counts | ZT0 Log$_2$ fold-change | ZT0 Log$_2$ s.e | ZT0 adj. p-value | ZT2 Mean read counts | ZT2 Log$_2$ fold-change | ZT2 Log$_2$ s.e | ZT2 adj. p-value | ZT4 Mean read counts | ZT4 Log$_2$ fold-change | ZT4 Log$_2$ s.e | ZT4 adj. p-value | Cluster | Membership |
|---|---|---|---|---|---|---|---|---|---|---|---|---|---|---|---|---|---|
| *Medtr4g070320* | *MtCBL2* | transcription factor | $3.90 \times 10^{-02}$ | 1200 | −0.80 | 0.15 | $1.60 \times 10^{-07}$ | **420** | **−1.40** | **0.16** | **$1.40 \times 10^{-17}$** | 1900 | −0.86 | 0.14 | **$1.50 \times 10^{-08}$** | 10 | 0.16 |
| *Medtr7g053410* | *MtCBL3* | BHLH transcription factor | $2.70 \times 10^{-05}$ | 150 | 0.08 | 0.13 | $6.40 \times 10^{-01}$ | 130 | −0.52 | 0.14 | **$5.30 \times 10^{-04}$** | 210 | 0.39 | 0.12 | **$2.70 \times 10^{-03}$** | 3 | 0.083 |
| *Medtr8g012290* | *MtCBL4* | BHLH transcription factor | $9.40 \times 10^{-02}$ | – | – | – | – | – | – | – | – | – | – | – | – | – | – |
| *Medtr1g017350* | *MtCBL5* | transcription factor | $1.70 \times 10^{-02}$ | 57 | 0.67 | 0.23 | **$7.00 \times 10^{-03}$** | 30 | −0.26 | 0.27 | $4.10 \times 10^{-01}$ | 54 | −0.24 | 0.23 | $3.60 \times 10^{-01}$ | 16 | 0.2 |
| *Medtr5g048860* | *MtCBL6* | BHLH transcription factor | $1.40 \times 10^{-01}$ | – | – | – | – | – | – | – | – | – | – | – | – | – | – |
| *Medtr8g065740* | *MtCBL7* | transcription factor | $8.00 \times 10^{-01}$ | – | – | – | – | – | – | – | – | – | – | – | – | – | – |
| *Medtr3g498825* | *MtCBL8* | transcription factor bHLH137 | $2.80 \times 10^{-12}$ | 920 | −0.04 | 0.07 | $6.30 \times 10^{-01}$ | 910 | −0.41 | 0.07 | **$7.80 \times 10^{-09}$** | 1000 | 0.32 | 0.07 | **$7.00 \times 10^{-06}$** | 18 | 0.18 |
| *Medtr8g099880* | *MtCBL9* | basic helix loop helix protein BHLH8 | $8.30 \times 10^{-02}$ | – | – | – | – | – | – | – | – | – | – | – | – | – | – |
| *Medtr8g062240* | *MtCBL10* | transcription factor | $6.00 \times 10^{-02}$ | – | – | – | – | – | – | – | – | – | – | – | – | – | – |
| *Medtr5g037250* | *MtCBL11* | transcription factor | $5.90 \times 10^{-14}$ | 170 | −0.57 | 0.18 | $3.10 \times 10^{-03}$ | **180** | **−1.10** | **0.18** | **$3.50 \times 10^{-09}$** | 70 | **1.10** | **0.21** | **$9.10 \times 10^{-07}$** | 18 | 0.5 |
| *Medtr7g092510* | *MtCBL12* | transcription factor | $3.50 \times 10^{-07}$ | 270 | 0.07 | 0.11 | $6.00 \times 10^{-01}$ | 210 | −0.65 | 0.12 | **$2.70 \times 10^{-07}$** | 360 | −0.78 | 0.11 | **$2.70 \times 10^{-12}$** | 11 | 0.3 |
| *Medtr1g059270* | *MtCBL13* | transcription factor | $2.60 \times 10^{-06}$ | 320 | 0.52 | 0.09 | $3.50 \times 10^{-08}$ | 320 | −0.14 | 0.09 | $1.60 \times 10^{-01}$ | 230 | 0.36 | 0.10 | **$6.40 \times 10^{-04}$** | 1 | 0.22 |
| *Medtr6g084120* | *MtCBL14* | transcription factor | $3.20 \times 10^{-11}$ | **92** | **−1.40** | **0.25** | **$1.20 \times 10^{-07}$** | **80** | **1.00** | **0.25** | **$8.40 \times 10^{-05}$** | 25 | **−1.10** | **0.32** | **$2.30 \times 10^{-03}$** | 18 | 0.27 |
| *Medtr7g099540* | *MtPIF1a* | transcription factor | $5.10 \times 10^{-25}$ | **1000** | **1.00** | **0.08** | **$9.20 \times 10^{-37}$** | 1300 | 0.13 | 0.08 | $1.40 \times 10^{-01}$ | 1200 | −0.12 | 0.08 | $1.70 \times 10^{-01}$ | 6 | 0.22 |
| *Medtr1g069155* | *MtPIF1b* | transcription factor | $9.00 \times 10^{-01}$ | – | – | – | – | – | – | – | – | – | – | – | – | – | – |
| *Medtr1g084980* | *MtPIF3a* | phytochrome-interacting factor 3.1 | $6.90 \times 10^{-16}$ | **1900** | **−2.70** | **0.22** | **$1.00 \times 10^{-31}$** | 1200 | −0.44 | 0.22 | $7.70 \times 10^{-02}$ | 300 | −0.09 | 0.23 | $7.60 \times 10^{-01}$ | 18 | 0.34 |
| *Medtr7g111320* | *MtPIF3b* | phytochrome-interacting factor 3.1 | $3.20 \times 10^{-01}$ | – | – | – | – | – | – | – | – | – | – | – | – | – | – |
| *Medtr7g110810* | *MtPIF6* | helix loop helix DNA-binding domain protein | $1.30 \times 10^{-04}$ | **18** | **−2.40** | **0.54** | **$1.90 \times 10^{-05}$** | 27 | 0.64 | 0.45 | $2.20 \times 10^{-01}$ | 12 | 0.08 | 0.52 | $9.00 \times 10^{-01}$ | 18 | 0.2 |
| *Medtr3g449770* | *MtPIF45* | transcription factor | $4.50 \times 10^{-32}$ | **2500** | **−2.00** | **0.11** | **$1.40 \times 10^{-69}$** | 2800 | −0.34 | 0.11 | **$4.20 \times 10^{-03}$** | 3200 | −0.28 | 0.11 | **$2.20 \times 10^{-02}$** | 12 | 0.42 |
| *Medtr7g039110* | *MtPIF78* | transcription factor | $1.20 \times 10^{-03}$ | 5000 | −0.40 | 0.09 | **$5.80 \times 10^{-05}$** | 3800 | −0.08 | 0.09 | $4.60 \times 10^{-01}$ | 3400 | 0.13 | 0.09 | $2.20 \times 10^{-01}$ | 5 | 0.43 |
| *Medtr1g019240* | *MtPIL* | helix loop helix DNA-binding domain protein | – | – | – | – | – | – | – | – | – | – | – | – | – | – | – |
| *Medtr5g017040* | *MtSPT* | helix loop helix DNA-binding domain protein | – | – | – | – | – | – | – | – | – | – | – | – | – | – | – |

**Notes.**

The genes listed in this table are loci known or hypothesised to participate in the photoperiod pathway in legumes along with homologues of the core components of the pathway in *A. thaliana*.

They include potential *FT* promoter binding genes compiled by *Ridge et al. (2016)* from which the naming of *MtTOE1a* to *MtSPT* derives.

Table depicts the the adjusted *p*-value for the interaction between time and condition. Note that if the adjusted *p*-value is significant each contrast between conditions at timepoints ZT0, ZT2 and ZT4 is also given to facilitate identifying where the patterns of expression diverge. Included in these results is the mean normalised read counts for the gene at this timepoint.

If the interaction term adjusted *p*-value is not significant contrasts are omitted.

In all cases it is the expression in LD relative to SD which is tested. In addition the cluster assignment and membership value are listed. Differentially expressed results are in bold using an $\alpha = 0.05$.

resulted in 9,427 of the 9,516 genes having differing expression between LD and SD at a minimum of one timepoint with 6,437, 4,511 and 6,159 for ZT0, ZT2 and ZT4 respectively (Fig. S2A). We consider only genes with >2-fold differences with >10 mean normalised reads as DE and there were 3,192 genes meeting this criteria at at least one timepoint. This corresponds to 2131, 1062 and 2168 DE genes at ZT0, ZT2 and ZT4 respectively. Full results are listed in Table S5. In these filtered lists of genes 4.9% of these genes differed in the three timepoints and 32% differed at two or more timepoints demonstrating how most genes have a single peak, predominantly at ZT0 or ZT4 (Fig. 4A). Notably, there were fewer genes DE at ZT2 than the other timepoints and those that did mostly differed at one or both of the other timepeoints too. Only 27% of the 1,062 genes DE at ZT2 are unique to the timepoint (compared to 58.4% for ZT0 and 51.6% for ZT4).

The interaction term was found to be significant in the majority (91/146; 62%) of the candidate genes associated with photoperiodic regulated flowering including 19/24 (79%) of the circadian clock and photoreceptor candidate genes and a striking 8/8 (100%) of the selected *AP2* class of genes (Table 1). Of this set of 91 genes, 90 were statistically different ($\alpha = 0.05$) at one or more timepoints and for 61 genes the difference at at least one timepoint was greater than >2-fold difference with >10 mean normalised reads. For example, a gene encoding a predicted core component of the core circadian clock, *MtLHY*, is expressed at ~4-fold higher levels at ZT0 in SD than LD but this situation is reversed at ZT4 when it is ~7.5-fold higher in LD than SD.

To get a broader view of the results, the mean abundances of all 9,516 genes which altered their pattern of expression were taken and $\log_2$ transformed, before being standardised to have a mean of zero and standard deviation of one. The standardised mean abundances were then clustered into 18 clusters using c-means clustering (Table S6; see Fig. S3A for cluster number optimisation) and visualised in Fig. 4B. This algorithm assigns a membership score (between 0 and 1) for each gene to each cluster describing the degree to which an individual observation belongs to a given cluster (see Fig. S4 for distribution of membership scores). A gene is then assigned to the cluster for which it has the highest membership. Cluster 3, which has 464 genes in it, was of particular interest as *MtFTb1* is present. Thus these genes have patterns of expression similar to *MtFTb1* and may be involved in regulating *MtFTb1* or involved in similar processes. Clusters 9, 11 and 17 (648, 598 and 538 genes respectively) are also of interest as they contain genes with an opposite expression pattern to cluster 3, some of which may thus be negative regulators of *MtFTb1* (e.g., *MtFTa2* is in cluster 11). Specifically, these three clusters have peaks of expression at ZT4 in SD which are not present in LDs (Fig. 4C).

With regard to candidate photoperiodic flowering time genes, alongside *MtFTb1* in cluster 3 there were only two genes from Table 1. These were *MtSOC1a* (*Medtr7g075870*) and a BHLH transcription factor gene called *MtCBL3* (*Medtr7g053410*). *MtSOC1a* is a downstream target of *MtFTa1* (*Medtr7g084970*) and demonstrated to affect flowering time (*Jaudal et al., 2018*). Another cluster 3 candidate flowering related gene *Medtr3g101520* encodes a B3 domain, as does *E1*, the most important locus in the photoperiod pathway of the tropical legume soybean (*Glycine max* (L.) Merr.; *Xia et al., 2012*). On the other hand, the predicted circadian clock-like genes *PRR37a, b* and *PRR59a-c* (*Matsushika et*

Peerj

**Table 2  Photoperiod induced changes in the magnitude of gene expression in candidate flowering time loci not observed to alter their pattern over time (see Table 1).**

| Gene identifier | Name | Description (Mt4.0 annotation) | Condition adj. p-value | ZT0 | | | | ZT2 | | | | ZT4 | | | | Cluster | Membership |
|---|---|---|---|---|---|---|---|---|---|---|---|---|---|---|---|---|---|
| | | | | Mean read count | Log$_2$ fold-change | Log$_2$ s.e | adj. p-value | Mean read count | Log$_2$ fold-change | Log$_2$ s.e | adj. p-value | Mean read count | Log$_2$ fold-change | Log$_2$ s.e | adj. p-value | | |
| *Medtr2g049520* | *MtPHYE* | phytochrome protein | – | – | – | – | – | – | – | – | – | – | – | – | – | – | – |
| *Medtr1g016920* | *MtELF3b* | EARLY flowering protein, putative | $5.50 \times 10^{-01}$ | – | – | – | – | – | – | – | – | – | – | – | – | – | – |
| *Medtr8g015470* | *ELF3-like* | hypothetical protein | – | – | – | – | – | – | – | – | – | – | – | – | – | – | – |
| *Medtr8g020200* | *ELF4-like* | early flowering protein | **$1.60 \times 10^{-20}$** | 100 | −0.88 | 0.18 | **$1.50 \times 10^{-05}$** | 98 | **−1.2** | **0.19** | **$7.40 \times 10^{-09}$** | 210 | −0.94 | 0.16 | **$1.70 \times 10^{-07}$** | 13 | 0.43 |
| *Medtr4g125590* | *ELF4-like* | early flowering protein | $1.60 \times 10^{-01}$ | – | – | – | – | – | – | – | – | – | – | – | – | – | – |
| *Medtr2g036510* | *MtZTL* | galactose oxidase/kelch repeat protein | $8.20 \times 10^{-01}$ | – | – | – | – | – | – | – | – | – | – | – | – | – | – |
| *Medtr7g084970* | *MtFTa1* | flowering locus protein T | **$1.80 \times 10^{-27}$** | **28** | **5.2** | **0.79** | **$1.20 \times 10^{-09}$** | **16** | **6.4** | **1.2** | **$1.90 \times 10^{-06}$** | **55** | **6.8** | **0.89** | **$2.10 \times 10^{-12}$** | 1 | 0.32 |
| *Medtr6g033040* | *MtFTa3* | flowering locus protein T | – | – | – | – | – | – | – | – | – | – | – | – | – | – | – |
| *Medtr7g006690* | *MtFTb2* | flowering locus protein T | $5.20 \times 10^{-01}$ | – | – | – | – | – | – | – | – | – | – | – | – | – | – |
| *Medtr2g461760* | *MtFULa* | MADS-box transcription factor | $2.90 \times 10^{-01}$ | – | – | – | – | – | – | – | – | – | – | – | – | – | – |
| *Medtr7g016630* | *MtFULc* | MADS-box transcription factor | – | – | – | – | – | – | – | – | – | – | – | – | – | – | – |
| *Medtr8g033250* | *MtSOC1b* | MADS-box transcription factor | – | – | – | – | – | – | – | – | – | – | – | – | – | – | – |
| *Medtr8g033220* | *MtSOC1c* | MADS-box transcription factor | $9.80 \times 10^{-01}$ | – | – | – | – | – | – | – | – | – | – | – | – | – | – |
| *Medtr2g016030* | *MtCDFa* | Dof domain zinc finger protein | $7.60 \times 10^{-01}$ | – | – | – | – | – | – | – | – | – | – | – | – | – | – |
| *Medtr5g041380* | *MtCDFd* | DOF domain, zinc finger protein | $4.40 \times 10^{-01}$ | – | – | – | – | – | – | – | – | – | – | – | – | – | – |
| *Medtr6g027450* | *MtCDFf* | Dof zinc finger DOF5.2-like protein | **$2.10 \times 10^{-09}$** | **12** | **3.1** | **0.67** | **$4.10 \times 10^{-05}$** | **9.8** | **1.8** | **0.57** | **$5.50 \times 10^{-03}$** | **2.7** | **3.7** | **1.3** | **$9.70 \times 10^{-03}$** | 2 | 0.53 |
| *Medtr7g086780* | *MtCDFg* | Dof zinc finger DOF5.2-like protein | $5.00 \times 10^{-02}$ | – | – | – | – | – | – | – | – | – | – | – | – | – | – |
| *Medtr8g044220* | *MtCDFh* | DOF-type zinc finger DNA-binding family protein | **$9.50 \times 10^{-05}$** | 65 | 0.93 | 0.21 | **$1.10 \times 10^{-04}$** | 60 | 0.49 | 0.21 | **$4.00 \times 10^{-02}$** | 63 | 0.26 | 0.21 | $2.80 \times 10^{-01}$ | 2 | 0.4 |
| *Medtr5g009530* | *SPA1-like* | ubiquitin ligase cop1, putative | **$1.90 \times 10^{-06}$** | 890 | −0.22 | 0.062 | **$1.40 \times 10^{-03}$** | 980 | −0.28 | 0.061 | **$3.50 \times 10^{-05}$** | 850 | −0.096 | 0.063 | $1.70 \times 10^{-01}$ | 18 | 0.52 |
| *Medtr2g085210* | *SPA1-like* | ubiquitin ligase cop1, putative | **$2.40 \times 10^{-03}$** | 4100 | 0.075 | 0.086 | $4.30 \times 10^{-01}$ | 4600 | 0.36 | 0.086 | **$1.80 \times 10^{-04}$** | 4700 | 0.18 | 0.086 | $6.20 \times 10^{-02}$ | 6 | 0.45 |
| *Medtr4g009840* | *TPL-like* | topless-like protein | **$9.00 \times 10^{-11}$** | 3200 | −0.35 | 0.11 | **$3.30 \times 10^{-03}$** | 3000 | −0.51 | 0.11 | **$1.40 \times 10^{-05}$** | 2000 | −0.49 | 0.11 | **$4.30 \times 10^{-05}$** | 18 | 0.88 |
| *Medtr4g114980* | *TPL-like* | topless-like protein | **$1.20 \times 10^{-05}$** | 840 | −0.25 | 0.14 | $9.90 \times 10^{-02}$ | 960 | −0.42 | 0.13 | **$6.30 \times 10^{-03}$** | 1200 | −0.59 | 0.13 | **$8.90 \times 10^{-05}$** | 9 | 0.57 |
| *Medtr2g435370* | *TPL-like* | transducin family protein/WD-40 repeat protein | – | – | – | – | – | – | – | – | – | – | – | – | – | – | – |
| *Medtr2g435440* | *TPL-like* | topless-like protein | – | – | – | – | – | – | – | – | – | – | – | – | – | – | – |
| *Medtr7g112460* | *TPL-like* | topless-like protein | **$1.60 \times 10^{-18}$** | 570 | −0.57 | 0.12 | **$1.20 \times 10^{-05}$** | 430 | −0.65 | 0.12 | **$1.40 \times 10^{-06}$** | 450 | −0.73 | 0.12 | **$3.40 \times 10^{-08}$** | 10 | 0.5 |
| *Medtr2g065670* | *TPL-like* | topless-like protein | $6.70 \times 10^{-01}$ | – | – | – | – | – | – | – | – | – | – | – | – | – | – |
| *Medtr2g435380* | *TPL-like* | topless-like protein | – | – | – | – | – | – | – | – | – | – | – | – | – | – | – |
| *Medtr1g012820* | *TPL-like* | topless-like protein | – | – | – | – | – | – | – | – | – | – | – | – | – | – | – |

**Table 2** (*continued*)

| Gene identifier | Name | Description (Mt4.0 annotation) | Condition adj. *p*-value | ZT0 Mean read count | ZT0 Log$_2$ fold-change | ZT0 Log$_2$ s.e | ZT0 adj. *p*-value | ZT2 Mean read count | ZT2 Log$_2$ fold-change | ZT2 Log$_2$ s.e | ZT2 adj. *p*-value | ZT4 Mean read count | ZT4 Log$_2$ fold-change | ZT4 Log$_2$ s.e | ZT4 adj. *p*-value | Cluster | Membership |
|---|---|---|---|---|---|---|---|---|---|---|---|---|---|---|---|---|---|
| *Medtr1g082660* | *NF-YC like* | nuclear transcription factor Y protein | **6.40 $\times 10^{-03}$** | 370 | −0.38 | 0.094 | **2.90 $\times 10^{-04}$** | 490 | −0.14 | 0.089 | 1.60 $\times 10^{-01}$ | 530 | −0.065 | 0.088 | 5.10 $\times 10^{-01}$ | 12 | 0.34 |
| *Medtr7g113680* | *NF-YC like* | nuclear transcription factor Y protein | **1.10 $\times 10^{-10}$** | 490 | 0.37 | 0.1 | **1.40 $\times 10^{-03}$** | 380 | 0.46 | 0.11 | **1.40 $\times 10^{-04}$** | 400 | 0.41 | 0.11 | **6.30 $\times 10^{-04}$** | 4 | 0.69 |
| *Medtr8g104190* | *MtCOLi* | zinc finger constans-like protein | **2.00 $\times 10^{-02}$** | 5800 | −0.032 | 0.07 | 6.90 $\times 10^{-01}$ | 4100 | −0.092 | 0.071 | 2.50 $\times 10^{-01}$ | 6100 | −0.29 | 0.07 | **1.80 $\times 10^{-04}$** | 15 | 0.28 |
| *Medtr5g072780* | *MtCMF3* | CCT motif protein | 6.10 $\times 10^{-01}$ | – | – | – | – | – | – | – | – | – | – | – | – | – | – |
| *Medtr3g100040* | *MtCMF5* | GATA transcription factor | **4.50 $\times 10^{-02}$** | 240 | −0.4 | 0.13 | **8.00 $\times 10^{-03}$** | 180 | −0.14 | 0.14 | 3.80 $\times 10^{-01}$ | 210 | −0.063 | 0.14 | 6.80 $\times 10^{-01}$ | 14 | 0.83 |
| *Medtr3g100050* | *MtCMF6* | GATA transcription factor | 1.20 $\times 10^{-01}$ | – | – | – | – | – | – | – | – | – | – | – | – | – | – |
| *Medtr5g066510* | *MtCMF7* | GATA transcription factor | 2.40 $\times 10^{-01}$ | – | – | – | – | – | – | – | – | – | – | – | – | – | – |
| *Medtr4g093730* | *MtCMF8* | GATA transcription factor | 1.70 $\times 10^{-01}$ | – | – | – | – | – | – | – | – | – | – | – | – | – | – |
| *Medtr4g061910* | *MtCMF11a* | CCT motif protein | – | – | – | – | – | – | – | – | – | – | – | – | – | – | – |
| *Medtr4g061823* | *MtCMF11b* | CCT motif protein | – | – | – | – | – | – | – | – | – | – | – | – | – | – | – |
| *Medtr2g096080* | *MtCMF12* | CCT motif protein | 1.50 $\times 10^{-01}$ | – | – | – | – | – | – | – | – | – | – | – | – | – | – |
| *Medtr8g098725* | *MtCMF13* | CCT motif protein | – | – | – | – | – | – | – | – | – | – | – | – | – | – | – |
| *Medtr5g010120* | *MtCMF14* | CCT motif protein | – | – | – | – | – | – | – | – | – | – | – | – | – | – | – |
| *Medtr2g068730* | *MtCMF15* | CCT motif protein | – | – | – | – | – | – | – | – | – | – | – | – | – | – | – |
| *Medtr1g044785* | *MtCMF17* | CCT motif protein | **1.80 $\times 10^{-02}$** | 2.3 | -2 | 1 | 8.70 $\times 10^{-02}$ | 4.5 | −1.6 | 0.76 | 6.80 $\times 10^{-02}$ | 13 | −0.51 | 0.54 | 4.00 $\times 10^{-01}$ | 16 | 0.54 |
| *Medtr4g008090* | *MtCMF18* | GATA transcription factor, putative | – | – | – | – | – | – | – | – | – | – | – | – | – | – | – |
| *Medtr3g113070* | CCT domain gene | salt tolerance-like protein | – | – | – | – | – | – | – | – | – | – | – | – | – | – | – |
| *Medtr4g071200* | CCT domain gene | salt tolerance-like protein | – | – | – | – | – | – | – | – | – | – | – | – | – | – | – |
| *Medtr8g012290* | *MtCBL4* | BHLH transcription factor | 1.40 $\times 10^{-01}$ | – | – | – | – | – | – | – | – | – | – | – | – | – | – |
| *Medtr5g048860* | *MtCBL6* | BHLH transcription factor | 4.20 $\times 10^{-01}$ | – | – | – | – | – | – | – | – | – | – | – | – | – | – |
| *Medtr8g065740* | *MtCBL7* | transcription factor | 6.20 $\times 10^{-01}$ | – | – | – | – | – | – | – | – | – | – | – | – | – | – |
| *Medtr8g099880* | *MtCBL9* | basic helix loop helix protein BHLH8 | 5.40 $\times 10^{-01}$ | – | – | – | – | – | – | – | – | – | – | – | – | – | – |
| *Medtr8g062240* | *MtCBL10* | transcription factor | **8.90 $\times 10^{-03}$** | 110 | 0.066 | 0.16 | 7.20 $\times 10^{-01}$ | 64 | 0.28 | 0.19 | 1.90 $\times 10^{-01}$ | 82 | 0.72 | 0.18 | **3.60 $\times 10^{-04}$** | 4 | 0.31 |
| *Medtr1g069155* | *MtPIF1b* | transcription factor | 9.30 $\times 10^{-01}$ | – | – | – | – | – | – | – | – | – | – | – | – | – | – |
| *Medtr7g111320* | *MtPIF3b* | phytochrome-interacting factor 3.1 | 5.80 $\times 10^{-01}$ | – | – | – | – | – | – | – | – | – | – | – | – | – | – |
| *Medtr1g019240* | *MtPIL* | helix loop helix DNA-binding domain protein | – | – | – | – | – | – | – | – | – | – | – | – | – | – | – |
| *Medtr5g017040* | *MtSPT* | helix loop helix DNA-binding domain protein | – | – | – | – | – | – | – | – | – | – | – | – | – | – | – |

**Notes.**

The genes listed in this table are loci known or hypothesised to participate in the photoperiod pathway in legumes along with homologues of the core components of the pathway in *A. thaliana*.

They include potential *FT* promoter binding genes compiled by *Ridge et al. (2016)* from which the naming of *MtCOLi* to *MtSPT* derives.

Table depicts the the adjusted *p*-value for the interaction between time and condition. Note that if the adjusted *p*-value is significant each contrast between conditions at timepoints ZT0, ZT2 and ZT4 is also given to facilitate identifying where the patterns of expression diverge. Included in these results is the mean normalised read counts for the gene at this timepoint.

If the interaction term adjusted *p*-value is not significant contrasts are omitted.

In all cases it is the expression in LD relative to SD which is tested. In addition the cluster assignment and membership value are listed. Differentially expressed results are in bold using an $\alpha = 0.05$.

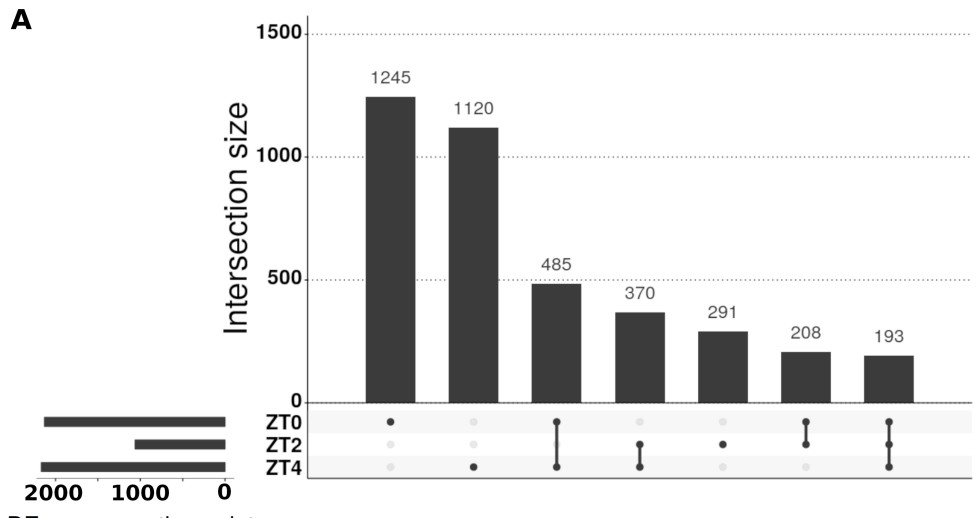

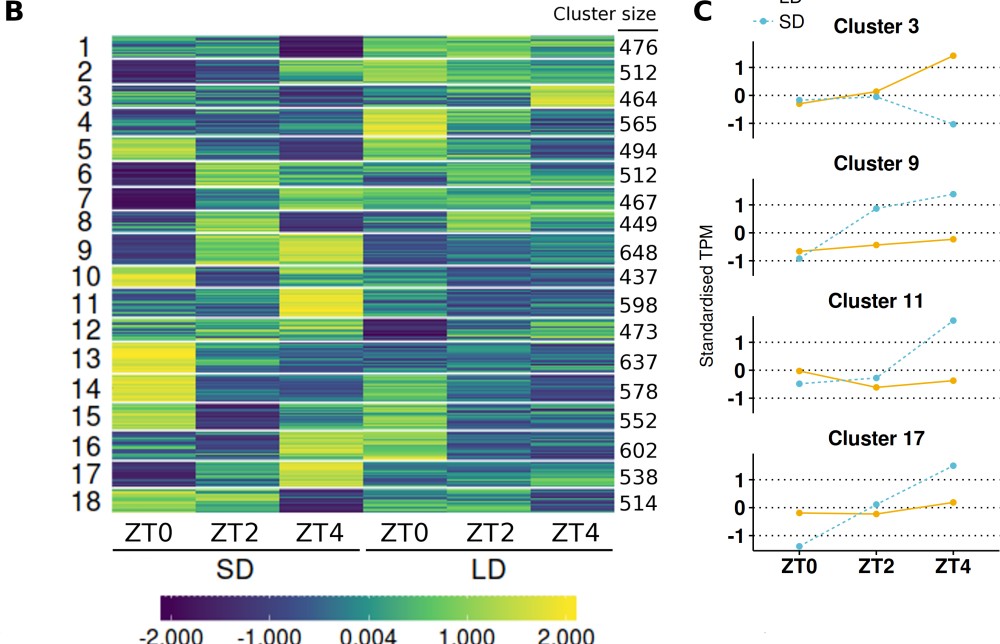

**Figure 4 Contrasting timepoint specific expression profiles of the genes which alter their pattern of gene expression in response to the change of photoperiod conditions and clustering the relative changes in expression over time.** (A) contrasts the 3,912 DE genes (>2-fold difference and >10 mean normalised reads) by the timepoints at which they are DE and plots their overlap. The principal chart plots the size of the overlaps and the supplementary chart presents the number of genes DE (both up or down) in LD relative to SD for each timepoint. (B) Heatmap of standardised gene abundances (such that the average expression value is zero and the standard deviation is one) of all 9,516 genes which alter their pattern of gene expression grouped into 18 clusters using c-means clustering. The number of genes in each cluster is listed alongside. (C) Mean standardised abundances for selected clusters. Specifically cluster 3 was selected as it contains *MtFTb1* which characteristically peaks at ZT4 in LD (orange and line) with no expression in SD (blue and dotted). Clusters 9, 11 and 17 were selected as their pattern is opposite to that of cluster 3 in that they peak at ZT4 in SD. Mean standardised abundances for all clusters are plotted in Fig. S5.

*al., 2000*; *Nakamichi et al., 2005*) all have profiles opposite to that of *MtFTb1* with three included in cluster 9 (Table 1). Additionally *Medtr1g033620*, a SHAQKYF class MYB-like DNA-binding domain gene, is present in cluster 11 which is interesting as proteins of this class have recently been implicated in the regulation of *FT* in *A. thaliana* (e.g., *EFM* and *FE*; *Yan et al., 2014*; *Abe et al., 2015*).

Next, genes in each cluster were then ranked by their fold-change at ZT4 because *MtFTb1* has its greatest difference in expression between the two conditions at this timepoint (962-fold up in LD; Fig. 1B). We focused on transcription factor genes and observed significant fold changes in transcript levels of genes encoding a number of zinc finger proteins. For instance, in cluster 3 at ZT4, the zinc finger (Ran-binding) family gene *Medtr4g113840* was 45-fold elevated in LD compared to SD and the DOF-type zinc finger DNA-binding family gene *Medtr3g091820* was increased 2-fold. In contrast, in cluster 9 the B-box type zinc finger protein *Medtr2g073370* was 29-fold more abundant in SD than in LD at ZT4 and in cluster 11, *Medtr2g059540*, also a DOF domain zinc finger gene (denoted *MtCDF1* in Table 1) was ~300-fold higher in SD than in LD. Furthermore in cluster 17, the zinc finger (Ran-binding) family gene *Medtr6g069400* is 7-fold elevated in SD compared to LD at ZT4.

### Changes in the magnitude of gene expression

There is a class of genes which change their level of expression in response to the shift from SD to LD conditions, but this does not alter the relative changes which occur at differing timepoints. Thus it is only the magnitude, not the pattern of expression which changes (e.g., *MtFTa1* or the *NF-YC-like* gene *Medtr1g082660* in Figs. 1A and 1G). To identify these genes a simpler model was fit the data which lacked the interaction term between growth condition and time of sampling.

It was observed that in this model 8,695 genes differed in the magnitude of their expression between conditions ($\alpha = 0.05$; Table S7), but this was reduced to 4,694 when those that also altered the pattern of their expression were omitted. Therefore 14.96% of genes with $>1$ read (4,694/31,363 genes) alter just the magnitude of their expression in response to the shift in photoperiod conditions. To provide timepoint level resolution of these changes Wald significance tests were again used to contrast the 4,694 genes (Table S8) with the significance levels of these genes adjusted for all three contrasts together using the false discovery rate method. This resulted in 4,161 of the 4,694 genes differing in magnitude at least one timepoint with 2,715, 2,268 and 2,666 for ZT0, ZT2 and ZT4 respectively (Fig. S2B). When considering genes with $>2$ fold differences and $>10$ mean normalised reads as DE, only 819 genes differed in magnitude at at least one timepoint and at ZT0, ZT2 and ZT4 this corresponded to 457, 353 and 454 genes respectively. Here it was observed that only 65 genes (7.9%) of this set of genes are consistently higher in LDs than SDs while 54 genes (9.99%) are consistently lower (Fig. 5A). The numbers in these classes go up to 187 genes (22.8%) and 139 genes (17%) respectively when genes differing at two or more timepoints are included. Like the class of genes which altered their pattern of expression there are fewer DE genes at ZT2.

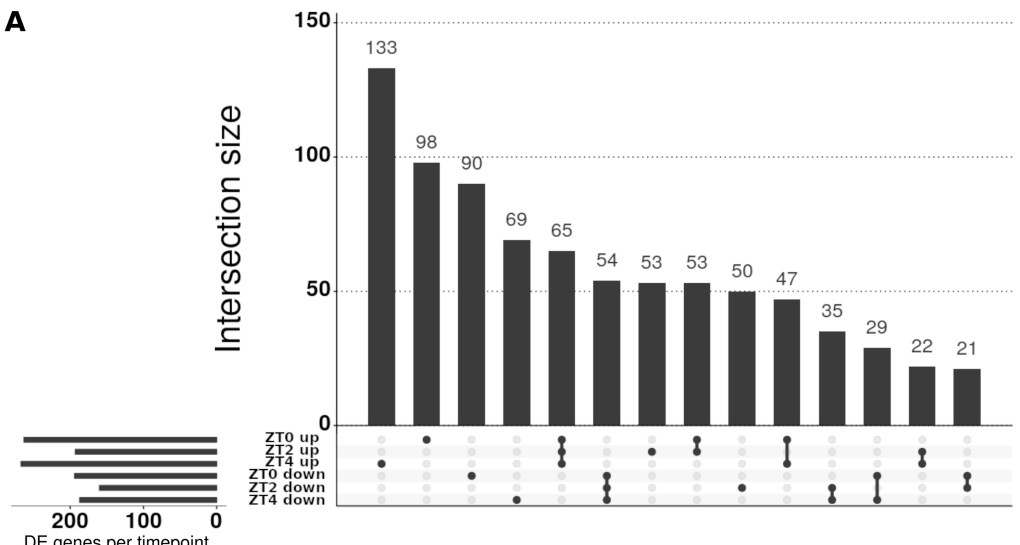

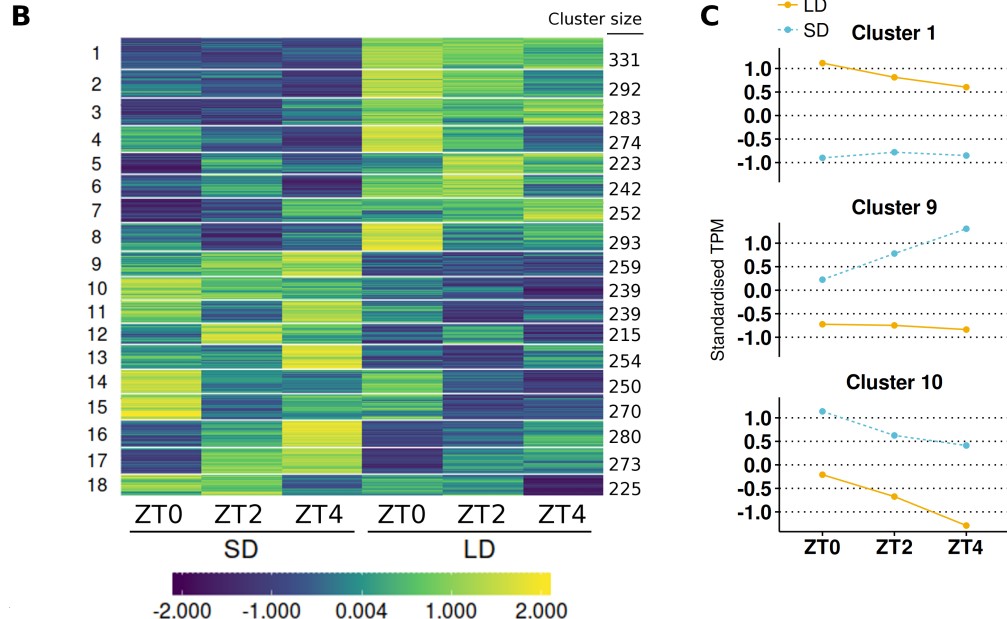

**Figure 5** **Contrasting timepoint specific expression profiles of the genes which alter only the magnitude of their gene expression in response to the change of photoperiod conditions and clustering the relative changes in expression over time.** (A) contrasts the direction of the 819 genes which alter just the magnitude of their gene expression and are DE (>2-fold difference and >10 mean normalised reads) at one or more timepoint. The principal chart plots the size of the overlaps between timepoints and the supplementary chart presents the number of genes DE (either up or down) in LD relative to SD for each timepoint. Membership within each group for individual genes, including non DE genes, is given in Table S8. (B) The standardised and clustered abundances (such that the average expression value is zero and the standard deviation is one) of all 4,694 genes which were classed as just altering the magnitude of their gene expression in response to the photoperiod shift clustered into 18 clusters using c-means clustering. The number of genes in each cluster is listed alongside. (C) Mean standardised abundance for clusters examined in more detail are plotted with cluster 1 selected for containing *MtFTa1* which is expressed in LD at all timepoints but not SD and clusters 9 and 10 were selected as genes in these clusters are consistently higher in SD (blue and dotted) than in LD (orange and line). Mean standardised abundances for all clusters are plotted in Fig. S6.

Results for the candidate photoperiod loci are summarised in Table 2. Specifically these are the candidate photoperiodic flowering time genes which were not classed as changing their pattern of expression between conditions. Here 15/55 (27%) of the genes are classed as altering just their magnitude in response to the photoperiodic shift. However, only 9/15 (60%) have statistically different levels of expression at two or more timepoints and looking down the list *MtFTa1* is the only gene to consistently differ >2-fold with >10 mean normalised reads. Thus, none of these candidate photoperiod genes are expressed similarly to *MtFTa1* which shows >30-fold higher levels in LD than in SD at all timepoints.

We then assessed the sets of genes whose expression is either consistently higher or lower in LDs than SDs (65 and 54 genes respectively) for other candidate genes which could potentially play a role in the regulation of flowering time. These include *Medtr8g091720* a *NF-Y-like* gene, which exhibits consistently greater expression (2.5–5.6-fold) in LD than in SD, like that of *MtFTa1*. Genes which consistently show reduced expression in LD compared to SD include *Medtr2g014200* which encodes a squamosa promoter-binding-like protein as well as genes associated with sugar transport. For instance, *Medtr3g074180* encoding a trehalose-6-phosphate phosphatase is 2–3 fold lower accross the three timepoints and *Medtr0204s0040* a sugar porter family MFS transporter which not expressed in LD at all. Sugar transport is a process linked to flowering time and the regulation of *FT* in *A. thaliana* (*Wahl et al., 2013*).

In addition, while not previously linked to flowering time, these lists also contain a number of genes which likely have regulatory functions such as *Medtr5g079220* encoding a R2R3-MYB transcription factor, *Medtr3g107940* which produces a FBD protein and *Medtr8g012655* which is the gene for an ethylene response factor. These genes all have greater expression in LD compared to SD. Conversely, genes which have reduced expression in LD compared to SD include *Medtr7g012790* encoding a circadian clock coupling factor ZGT, *Medtr7g105780* encoding an ovate transcriptional repressor, *Medtr3g031220* which produces a WRKY transcription factor and *Medtr8g026960* which is the gene for a homeobox associated leucine zipper protein.

Relaxing the criteria for differential expression slightly and reconsidering the candidate photoperiod loci also suggests that *MtCDFf* (*Medtr6g027450*) could be investigated further as it is higher in LD in all three timepoints (8.57-fold, 3.48-fold and 12.99-fold respectively) however its expression is overall quite low with an average of only 2.7 normalised reads at ZT4. Other potential candidates on the list include the *ELF4-like* gene *Medtr8g020200*, consistently 1.8–2.3-fold higher in SD compared to LD, as is *MtCMF17* (*Medtr1g044785*) although, like *MtCDFf*, the expression in these datasets is low. This may reflect that these genes are cell-type specific and so only expressed in a fraction of those sampled.

All 4,694 genes which differed in the magnitude of their expression between conditions were then clustered into 18 clusters (Fig. 5B, Fig. S3B and Table S9). *MtFTa1* was present in cluster 1 which had 331 genes. However, none of the selected photoperiodic candidate genes clustered with *MtFTa1*. Clusters 9 (259 genes) and 10 (239 genes) have patterns opposite to that of cluster 1 (Fig. 5C). A *TPL-like* gene from Table 2 is in cluster 9 and a second *TPL-like* gene is in cluster 10. In addition, in clusters 9 and 10, a number of other flowering time candidates are present including *Medtr0020s0120* in cluster 9, which is similar to

the *FT* antagonist *TERMINAL FLOWER 1* in *A. thaliana* (*Jaeger et al., 2013*; *Wickland & Hanzawa, 2015*). Also present in cluster 9 are a trio of genes encoding B3 binding domain proteins *Medtr1g021410*, *Medtr1g021435* and *Medtr1g021500* and pair of genes which encode SHAQKYF class MYB transcription factors *Medtr0036s0260* and *Medtr5g027550*. These genes are notable for genes containing these domains have been associated with flowering time, as has the jumonji domain protein encoding gene *Medtr2g011630* in cluster 10 (*Xia et al., 2012*; *Abe et al., 2015*; *Yan et al., 2014*).

A total of 35/65 of the consistently differentially expressed genes which are higher in LDs than SDs (Fig. 5A) are present in cluster 1. These include *Medtr1g099440* which encodes a membrane-associated kinase regulator-like protein, *Medtr6g086805* a heat shock transcription factor gene and *Medtr4g009110* which encodes a helix loop helix DNA-binding domain protein. Similarly, clusters 9 and 10 contain 19/54 of the consistently differentially expressed genes which are lower in LDs than SDs. These genes include the ethylene response factor gene *Medtr5g016750* and *Medtr4g119422* encoding a cullin-like protein.

We then ranked the clustered genes by their fold-change between LD and SD at ZT0. Strikingly, in cluster 1, within the top 13 genes with the largest fold increase in LD compared to SD at ZT0, *MtFTa1* is ranked 8th, while 10 of the other genes were homologues of *IRON MAN* (*IMA*)/*FE-UPTAKE-INDUCING PEPTIDE 1* (*FEP1*) genes. These encode mobile signalling peptides integral for the uptake of of iron from the soil and octuple *ima*/*fep1* mutants in *A. thaliana* result in severe chlorosis (*Grillet et al., 2018*; *Hirayama et al., 2018*). They were also identified in a recent reannotation of the *M. truncatula* genome to identify small, secreted peptides (*De Bang et al., 2017*). In addition cluster 3, which is similar to cluster 1, has the remaining annotated *IMA*/*FEP1* genes as 4 of the top 5 genes with the largest fold-change differences at ZT0.

## DISCUSSION

This study presents a thorough overview of the changes in the *M. truncatula* leaf tissue transcriptome following a shift of vernalised plants from SD to LD conditions between ZT0 and ZT4. Our data are of very high quality with an average mapping rate of 89.86%. This suggests that while the existing Mt4.0v2 transcriptome captures the majority of the signal in the data there is nevertheless space to improve it. To date, a majority of transcriptomic datasets in *M. truncatula* have been generated from root tissue, not leaves (e.g., experiments in the MtGEA database are >50% root tissue (*Benedito et al., 2008*)). Thus this dataset broadens the understanding of gene expression in the aerial tissue of *M. truncatula*, in particular in differing photoperiodic conditions. Consequently it could be incorporated into future cross-species comparisons with datasets like that of *Wu et al. (2014)* who performed a similar shift experiment in the SD-responsive soybean.

While the data presented here consist of two composite datasets, and so interpretation requires caution, the results of this analysis are nevertheless biologically plausible. Initial pairwise comparisons of gene expression between LD and SD at each timepoint individually qualitatively agreed with previously published expression profiles of *MtFTa1*, *MtFTb1*

and *MtFTb2* (Figs. 1A–1C; *Laurie et al., 2011*). Furthermore, independent RT-qPCR timecourses of *MtFKF1*, *MtCDF1*, *MtCDF2* and *MtCDF4* are similar to the transcript abundances seen in these RNA-Seq datasets (Fig. 2). From these results it was concluded that there is no significant batch effect between datasets and that it is appropriate to interpret this data as a time series.

The dual approaches taken to analyse the data as a time series first considered the interaction between condition and time and then just the effect of the condition. This successfully identified the genes which alter the pattern or just the magnitude of their expression respectively. This approach could serve as a template for similar datasets in other plants which lack a CO-like regulator. Given the significant role the circadian clock plays in the regulation of the *A. thaliana* transcriptome (*Covington et al., 2008*; *Michael et al., 2008*) and the manner in which the photoperiod regulates the circadian clock (*Nohales & Kay, 2016*), it is unsurprising to see that a greater number of genes were classed as altering their pattern of expression (9,516/31,363 of detectable genes; 30.34%) than only the magnitude of their expression (4,694/31,363; 14.96%). This is especially true of our selected candidate photoperiodic genes (Table 1) where 62% had a significant change in pattern.

Clustering was employed to subset the two classes of DE genes further based on their normalised abundances across the three timepoints. The low membership scores reveal the small degree of separation between clusters. This may be a feature of gene expression data, but the strength of c-means clustering is that it allows the certainty of cluster assignment to be assessed (Tables S6, S9). An alternative approach to subset the classes of genes would be to group them based on functional gene set descriptors such as Gene Ontology (GO) terms and cluster until individual clusters become enriched for single GO terms. However since in *M. truncatula* only 37% of genes have annotated GO terms (*Tang et al., 2014*), their use is currently of limited utility.

This study has identified additional candidate photoperiodic flowering time genes for future characterising and reverse genetics screens. In terms of identifying genes co-expressed with the LD-induced *M. truncatula FT-like* genes, candidate genes such as the zinc finger gene *Medtr4g113840* and the B3 domain transcription factor gene *Medtr3g101520* (both in the same cluster as *MtFTb1*) present as future avenues of inquiry as potential regulators of photoperiodic flowering. Conversely, it would be interesting to investigate the CCT containing B-box type zinc finger gene *Medtr2g073370* which has the opposite pattern of expression to *MtFTb1* consistent with a repressive role. Finally, it is notable that none of the list of candidate photoperiod genes responded in a similar way to the SD to LD photoperiodic shift as the potent floral activator *MtFTa1* (Table 2). However a number of other potential regulators were identified, such as the *NF-Y-like Medtr8g091720* which has consistently higher expression in LD or the pair of *TPL-like* genes (*Medtr4g114980* and *Medtr7g112460*) and ethylene response factor gene *Medtr5g016750* which all have a pattern of expression opposite to *MtFTa1*.

The experiments analysed in this study focused on the first four hours of the diurnal cycle where, in LD, *MtFTa1* is induced and which precede and include the first peak in expression of LD induced *MtFTb1* and *MtFTb2* at ZT4 (*Laurie et al., 2011*). It should be

noted that the clustering presented in this study is limited to the timepoints sampled and cannot be considered predictive of the pattern in which genes are expressed later in the day. Genes in the same cluster may have divergent patterns later in the day. Given the diurnal pattern of expression of *MtFTb1* and *MtFTb2* (*Laurie et al., 2011*), our ability to identify candidate regulators which share patterns of expression would be enhanced by the inclusion of additional samples from later timepoints. Notably at ZT8 to capture the trough and ZT16 to capture the second peak in expression of *MtFTb1* and *MtFTb2* in LD. This is because samples from later timepoints would facilitate greater discrimination between expressed genes and thus result in smaller clusters. However, it is also possible that the regulator of *M. truncatula FT-like* gene expression is post-translationally regulated because light/protein dependent mechanisms are common in photoperiodic and circadian regulatory networks. In this case, it might not be possible to identify the regulators using a co-expression approach.

## CONCLUSIONS

This study further elucidates the photoperiodic acceleration of flowering in the reference legume species *M. truncatula* which interestingly appears to lack a CO-like regulator. We found that the photoperiodic shift from SD to LD conditions had a large effect on the leaf transcriptome with 14,210 genes altering their pattern or magnitude of expression. Candidate regulators that were co-expressed with the LD-induced *FT-like* genes were identified by clustering. It was notable that none of the list of candidate photoperiod genes responded to the photoperiodic shift in a similar manner as that of the potent floral activator *MtFTa1*, and few were similar to that of the *MtFTb* genes. Thus this analysis further supports the idea that *FT-like* genes in *M. truncatula* are uncoupled from the photoperiodic transcriptional networks seen in other species and that flowering time in *M. truncatula* is induced in a novel manner. Future work will focus on molecular-genetic analysis of the function of the candidate regulators identified in this study in *M. truncatula* photoperiodic flowering.

## ACKNOWLEDGEMENTS

We would like to thank our anonymous reviewers, Nicole Cloonan, Peter Tsai, Kevin Chang and William Schierding for their advice and insight in analyzing the data. Many thanks also to the other members of the Plant Molecular Biology Lab at the University of Auckland, especially Betty Phan and Lulu Zhang.

### Funding

The research was funded by C10X0816 MeriNET (www.msi.govt.nz/) and by the New Zealand Marsden Fund (http://www.royalsociety.org.nz/programmes/funds/marsden/) contract 14-UOA-125. The funders had no role in study design, data collection and analysis, decision to publish, or preparation of the manuscript.

## Grant Disclosures

The following grant information was disclosed by the authors:
C10X0816 MeriNET.
New Zealand Marsden Fund: contract 14-UOA-125.

## Competing Interests

The authors declare there are no competing interests.

## Author Contributions

- Geoffrey Thomson conceived and designed the experiments, performed the experiments, analyzed the data, contributed reagents/materials/analysis tools, prepared figures and/or tables, authored or reviewed drafts of the paper, approved the final draft.
- James Taylor performed the experiments, contributed reagents/materials/analysis tools, approved the final draft.
- Joanna Putterill conceived and designed the experiments, analyzed the data, contributed reagents/materials/analysis tools, prepared figures and/or tables, authored or reviewed drafts of the paper, approved the final draft.

## Ethics

The following information was supplied relating to ethical approvals (i.e., approving body and any reference numbers):

The University of Auckland Institutional Biological Safety Committee granted Biological Safety Approval to carry out this study within its facilities (GMO08-UA006).

## Data Availability

Raw data is available at GEO, accession number: GSE118893.

Code is available at Figshare: https://doi.org/10.17608/k6.auckland.6993641.v5.

## Supplemental Information

Supplemental information for this article can be found online at http://dx.doi.org/10.7717/peerj.6626#supplemental-information.

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
