# Peer review of "The transcriptomic response to a short day to long day shift in leaves of the reference legume Medicago truncatula"

_PeerJ, doi:10.7717/peerj.6626_

## Round 0.1 · original submission · Minor Revisions

Please address all the comments of the reviewers.

It was suggested to add expression data from a couple transcripts from cluster 1 (LD/SD time-course) into the supplement, as a proof of concept.

The reviewers found several typos - it should be easy to fix them.

Reviewer 1 ·

Basic reporting

The paper describing a brief overview of the changes in the M. truncatula leaf tissue transcriptome following a shift of vernalised plants is well-written and presented study. It is a well written paper with minor typos .

Experimental design

The experimental designs are presented well with justification and analysis of each finding given with the paper.

Validity of the findings

The tables, and figures are documented well to justify the experimental design and results. It was a nice study and results to read. The authors can possibly combine the result and discussion part to make it robust or expand the discussion slightly as the result section contains parts from discussion.

Reviewer 2 ·

Basic reporting

The study is well written overall and the results are clearly presented.

Experimental design

The authors have performed a straightforward transcripomic analysis of plants moved from SD to LD. The only slightly confusing aspect is that they have combined two different shift experiments on plants of different ages into one analysis, but they appear to have controlled for this.

Validity of the findings

The findings appear reasonable. The authors confirm induction of known flowering genes, suggesting that the experiment has worked and the results are meaningful.

Additional comments

The transcriptomic response to a short day to long day shift in leaves of the reference legume Medicago truncatula

In this study, Thomson et al investigate the response of Medicago leaves to inductive long days using an RNA-seq approach. In two experiments, Medicago plants are shifted from SD to LD conditions, and the transcriptome assayed at ZT0, 2 and 4 h.

The authors analyse the transcriptome data bioinformatically to determine if they can identify transcriptional responses likely to be indicative of the pathway that controls flowering in Medicago. While FT has been identified to be conserved in Medicago, the pathways controlling the induction of flowering in Medicago are much less well understood. This is an interesting study, and the authors have done a sound job of carrying out the experiments. The following are some suggestions for the study:

(1) Line 247 discusses the difficulty of knowing which timepoint is most relevant to the regulation of FT-like genes. Did the authors consider using the expression of FT as a marker for co-expression. A transcriptional positive regulator of FT that itself is transcriptionally regulated would then be expected to have the same pattern of expression of FT. This might be a straightforward way to identify candidate FT regulators. (this is analogous to the clustering used in Fig 4b, but more direct, the authors might consider this to be sufficient for this purpose.)
(2) An implicit assumption of the analysis and discussion is that the regulator of FT-like genes in Medicago is transcriptionally regulated. In this way, the authors find some interesting candidate TFs that are co-expressed with FT, and are thus good candidates. Another possibility is that the regulator of FT expression is post-translationally regulated. The authors should mention this specifically in the discussion, because it is important to bear in mind when considering the conclusions.
(3) The experimental setup is not completely clear. For the plants grown in SD, at what ZT time where they shifted to LD? How many hours/days were they grown in LD before the next sampling? It seems that the LD shifted plants were sampled after exposure to a single LD (or shortened night), is this correct? (Is it known for example if a single LD is sufficient to induce flowering in Medicago? The plant may have a more robust photoperiodic response if it is entrained for several days in LD. In any case it would be helpful to discuss this)


Minor:
Line 367: “more than >2-fold” is redundant.

Reviewer 3 ·

Basic reporting

The manuscript is well written in standard English with sufficient background information provided. The presentation and display are of high quality.

Experimental design

The research falls in the scope of the journal and the focused question is well defined and meaningful. Methods are thoroughly described.

Validity of the findings

The conclusion is well supported by the data presented and the data are statistically sound.

Additional comments

The manuscript submitted by Geoffrey Thomson et al. systematically explored the transcriptomic datasets associated with photoperiodic flowering when Medicago truncatula plants are shifted from short day (SD) to long day (LD). Though flowering has been studied extensively in the model plant species Arabidopsis, legume species seem to have some unique mechanisms that diverge from Arabidopsis. It is worth deciphering the details of these mechanisms. As a model legume, M. truncatula is an ideal material for molecular mechanistic investigation in photoperiodic flowering. Overall, the submitted manuscript sets up a solid foundation for future studies by systematically analyzing the transcriptomic data during the SD to LD shift. The manuscript is well written and the experiments were well designed. I only spotted a few typos to be changed:
Line 56, “suggest” should be “suggests”;
Line 98, delete “it” in “M. truncatula it is”;
Line 132, delete one “was” in “Soil was was kept”;
Line 145, change “was” to “were”;
Line 147, change “prepared” to “were prepared”;
Line 156, change “is run” to “was run”;
Line 202, change “is of a high” to “are of a high”;
Line 248, change “genes abundances” to “gene abundances”;
Line 292, delete “genes” in the phrase “criteria genes at at least”;
Line 301, change “strikingly” to “striking”;
Line 383, delete “are”;
Line 385, change “a ovate” to “an ovate”;
Line 400, change “were” to “are”;
Line 403, change “code” to “encode”;
Line 424, change “is ” to “are”;
Line 433, change “consists” to “consist”.

Reviewer 4 ·

Basic reporting

My first comment pertains to the paragraph starting line 53 of the introduction. I think this paragraph emphasizes well the difference between the universality of FT-dependent regulation in flowering plants, contrasted against the pathways to which FT expression is coupled. While I think this is a fascinating evolutionary question, I feel it could be phrased more succinctly, and it is a number of paragraphs until you make the point that in Medicago, CO-dependent regulation of FT is not used. I would considering re-ordering this paragraph or moving segments of it to the later paragraph to better hone in on this point. For instance, as a suggestion, you might use this to say CO-dependent regulation is a common regulatory mechanism in species X,Y,Z which could be a convergent coupling to FT regulation, but is not universal, and cite a few extra examples. Because the paper is partially framed around helping find the mystery MtFt1a regulator, I think special care in these two paragraphs will help frame the central biological question better.

Small points:

Figure 2A: Left most panel, right side has been cut off slightly

Figure 3: Please list what PCA1,2 are in the figure legend (sampling time and photoperiod)

Two small typos on lines 382 and 283.

Experimental design

Most of my comments on experimental design relate to time point choice and time point utility using the approach outlined in this study by the authors:

Paragraph beginning line 196/ also final paragraph of introduction line 118: While I think the authors are very clear throughout in their experimental methods, I would really like to see a more detailed discussion in either section as to why the time-points ZT0,2, and 4 were chosen (This is mentioned in the abstract, and I’d like to see it again in the results/ or at the end of the intro in greater detail). While these time-points seem sufficient for capturing upregulation of MtFTa1 expression after LD treatment, and coincides with the initial peak of the MtFTb genes under LD, and probably similar co-expressed genes, I was curious as to why a more dispersed time series was not investigated (i.e. 0, 4, 8, for example which would capture MtFTb morning peak and trough), and I think that the author’s commentary on time point choice would be valuable for future readers who are planning a similar analysis.

Similarly, I think this study hinges on its utility of being able to identify co-expressed transcripts with similar diurnal and photoperiodic behavior to MtFTa1/ MtFTb. While I completely agree that the authors’ analysis captures sufficiently the expression of photoperiodic genes over the time points used, as highlighted in Figure 1, I was curious if the span of time-points and kinetics for genes in this cluster are able to predict behavior at other times in the day in any meaningful way (especially for the particularly unique expression of MtFT1a). If RNA/cDNA is still available from the time courses in Figure 1, I think it might be helpful to add expression data from a couple transcripts from cluster 1 (LD/SD time-course) into the supplement, as a proof of concept (even though they are perhaps not involved in the transition to flowering). This might include Medtr1g099440, Medtr6g086805, the IRON MAN (IMA) transcripts, or Medtr4g009110.
If the authors think the clustering analysis over their time-points is not predictive in this way, it might be useful to add a short segment about this in the discussion, and how one might design a future study to capture co-expressing transcripts for a given photoperiodic gene of interest. I think this would be of utility for others who are interested in using this approach.

Validity of the findings

In my own experience, the primary draw back of utilizing a co-expression approach for finding, for instance, an MtFT1a regulator, is that one could imagine a constitutively expressed factor whose activity is temporally regulated by light, either through interaction with a photoreceptor/photoreceptor coupled pathway or through light dependent protein level degradation. In either case, you potentially might not be able identify the regulator using this approach. Perhaps if the regulator activated or repressed several other genes at the same level of MtFT1a, for instance, then one may have enough information to investigate candidates based on shared promoter regulatory elements, etc. Considering that these light/protein dependent mechanisms are common in photoperiodic and circadian regulatory networks, I think it is worth discussing this drawback of using the above approach briefly in the beginning of the discussion.

Additional comments

The effective use of RNA-seq data sets to the study of photoperiodic pathways poses some unique challenges in comparison to more linear signaling pathways, and this study provides a great resource to help tackle some of those challenges. I commend the authors on a very well written and assembled study, and think that it will provide a great resource for others studying photoperiodic flowering, as well as seasonally regulated pathways that are less intensively studied compared to flowering induction.

---

## Round 0.2 · accepted · Accept

Thank you very much for improving the manuscript. You paper will be of interest for a wide range of plant researchers.

#